# HOW EFFICIENT IS LLM-GENERATED CODE?
# A RIGOROUS & HIGH-STANDARD BENCHMARK

**Ruizhong Qiu**[†], **Weiliang Will Zeng**[‡], **James Ezick**[‡], **Christopher Lott**[‡] **& Hanghang Tong**[†]

[†]University of Illinois Urbana–Champaign     [‡]Qualcomm AI Research

{rq5,htong}@illinois.edu    {wzeng,jezick,clott}@qti.qualcomm.com

## ABSTRACT

The emergence of large language models (LLMs) has significantly pushed the frontiers of program synthesis. Advancement of LLM-based program synthesis calls for a thorough evaluation of LLM-generated code. Most evaluation frameworks focus on the (functional) correctness of generated code; efficiency, as an important measure of code quality, has been overlooked in existing evaluations. In this work, we develop ENAMEL (EfficeNcy AutoMatic EvaLuator), a rigorous and high-standard benchmark for evaluating the capability of LLMs in generating efficient code. Firstly, we propose a new efficiency metric called eff@$k$, which generalizes the pass@$k$ metric from correctness to efficiency and appropriately handles right-censored execution time. Furthermore, we derive an unbiased and variance-reduced estimator of eff@$k$ via Rao–Blackwellization; we also provide a numerically stable implementation for the new estimator. Secondly, to set a high standard for efficiency evaluation, we employ a human expert to design best algorithms and implementations as our reference solutions of efficiency, many of which are much more efficient than existing canonical solutions in HumanEval and HumanEval+. Moreover, to ensure a rigorous evaluation, we employ a human expert to curate strong test case generators to filter out wrong code and differentiate suboptimal algorithms. An extensive study across 30 popular LLMs using our benchmark ENAMEL shows that LLMs still fall short of generating expert-level efficient code. Using two subsets of our problem set, we demonstrate that such deficiency is because current LLMs struggle in designing advanced algorithms and are barely aware of implementation optimization. Our benchmark is publicly available at https://github.com/q-rz/enamel.

## 1 INTRODUCTION

The emergence of large language models (LLMs; Brown et al., 2020; Touvron et al., 2023) has driven the frontiers of program synthesis (Simon, 1963; Gulwani et al., 2017) with the help of large open codebases for pretraining. A number of code LLMs have been released (Chen et al., 2021; Li et al., 2022; Nijkamp et al., 2023; Roziere et al., 2023). They autoregressively generate code from a prompt that describes the requirement (e.g., in the form of a function signature and a docstring). Advancement of LLM-based program synthesis in turn calls for a thorough evaluation of LLM-generated code. Most of the existing evaluation frameworks (Chen et al., 2021; Austin et al., 2021; Hendrycks et al., 2021; Cassano et al., 2022; Lai et al., 2023; Liu et al., 2023a) focus on the *(functional) correctness* of generated code. Each framework has a collection of programming problems along with test cases, which are used to evaluate the correctness of generated codes.

Apart from correctness, however, *efficiency* is another important measure of code quality and has been overlooked in existing evaluations. Code efficiency is crucial in real-world applications for boosting system throughput, improving algorithm latency, and reducing energy consumption. Nonetheless, not until very recently have a few benchmarks (Nichols et al., 2024; Niu et al., 2024; Huang et al., 2024; Du et al., 2024) been proposed to evaluate the efficiency of LLM-generated code, and a number of fundamental challenges remain uncharted and open:

Qualcomm AI Research is an initiative of Qualcomm Technologies, Inc. All datasets were downloaded and evaluated at the University of Illinois Urbana–Champaign.

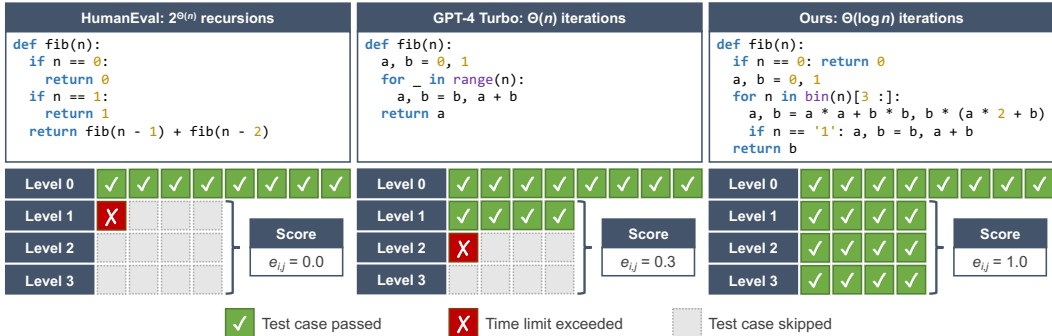

Figure 1: Illustration of our ENAMEL framework with HumanEval problem #55 (computing the $n$-th Fibonacci number). Our level-based evaluation clearly differentiates the three algorithms: (i) a naïve algorithm that needs $2^{\Theta(n)}$ recursions, (ii) a dynamic programming algorithm that needs $\Theta(n)$ iterations, and (iii) an efficient doubling algorithm that needs only $\Theta(\log n)$ iterations.

**(C1) Right-censored execution time.** When code execution is early terminated due to time limit, its actual execution time is unknown; this is *right censoring* in statistics (Bang & Tsiatis, 2000). For instance, if the generated code contains an infinite loop, the right-censored execution time will be clipped to the time limit while the actual execution time should be infinity. Existing works (Niu et al., 2024; Huang et al., 2024) use the execution time without coping with right censoring and thus *overestimate* the efficiency.

**(C2) Efficiency v.s. sample size.** Different code samples generated from LLMs for the same problem could have different execution times. We generalize the pass@$k$ metric (Chen et al., 2021) to characterize the efficiency given sample sizes $k$. Existing work either uses only one code sample (Niu et al., 2024) or averages the efficiency scores of code samples (Huang et al., 2024; Du et al., 2024); therefore, they fall short in capturing the relationship between code efficiency and the sample size $k$.

**(C3) Algorithm design & implementation optimization.** A good reference of efficiency should be the most efficient code, which often needs advanced algorithms and implementation optimization that can be highly non-trivial even for human programmers. Prior works either use existing canonical solutions provided in the dataset as the reference (Niu et al., 2024; Huang et al., 2024) or use solutions collected online (Du et al., 2024), but our evaluation reveals that many of the non-expert solutions themselves are inefficient and thus are not suitable references for efficiency.

**(C4) Correctness filter.** Wrong code can be efficient, but such code is useless. For example, an efficient yet wrong algorithm for deciding the primality of an integer is the Fermat primality test, which is known to have nontrivial counterexamples (Carmichael, 1912). Thus, we need to use strong test cases to filter out wrong code and evaluate efficiency only with correct code. Niu et al. (2024) rely on existing test cases provided by the dataset, but Liu et al. (2023a) have shown that those tests are not strong enough to fully detect wrong code.

**(C5) Worst-case efficiency.** Some suboptimal algorithms can appear efficient on random inputs despite their inefficiency on strong inputs. For example, if we search for a length-$m$ substring in a length-$n$ string, a brute-force algorithm takes only $\Theta(n + m)$ time on random strings but requires $\Theta(nm)$ time in the worst case. Huang et al. (2024) and Du et al. (2024) use GPT to produce test case generators, but we found that their test cases are mostly random and thus cannot differentiate such suboptimal algorithms.

To collectively address the aforementioned challenges, we develop ENAMEL (EfficieNcy AutoMatic EvaLuator), a high-quality benchmark to rigorously evaluate the capability of LLMs in generating efficient code. We carefully select 142 problems out of the 164 problems in HumanEval (Chen et al., 2021) and HumanEval+ (Liu et al., 2023a), excluding trivial problems with $\Theta(1)$ time complexity. With a wide spectrum of easy to hard problems, we are able to comprehensively evaluate how capable the LLM is to generate efficient code for various problems. Our main contributions are as follows:

- **Efficiency metric & its unbiased, variance-reduced estimator.** We propose a new efficiency metric called eff@$k$, which generalizes the pass@$k$ metric from correctness to efficiency. Our eff@$k$ metric properly handles right-censored execution time (C1) and precisely characterizes the efficiency under different sample sizes $k$ (C2). Furthermore, we derive an unbiased, variance-reduced estimator of our eff@$k$ via Rao–Blackwellization, and provide a numerically stable implementation of our estimator.

- **Efficient reference solutions.** To set a high-standard for efficiency evaluation, we employ a human expert to design best algorithms and implementations as our reference solutions of efficiency (C3). Many of our reference solutions are much more efficient than the canonical solutions in HumanEval and HumanEval+. For example, the canonical solution of computing the $n$-th Fibonacci number in HumanEval+ needs $\Theta(n)$ iterations while our reference solution needs only $\Theta(\log n)$ iterations.

- **Strong test case generators.** To ensure a rigorous evaluation, we employ a human expert to curate strong test case generators that cover both corner cases to filter out wrong code (C4) and worst cases to differentiate suboptimal algorithms (C5). Under our generated strong test cases, 11 canonical solutions in HumanEval and 4 in HumanEval+ are found wrong, and 34 in HumanEval and 27 in HumanEval+ exceed the time limit.

- **Rigorous & high-standard benchmark.** We open-source ENAMEL, a rigorous and high-standard benchmark for evaluating the capability of LLMs in generating efficient code. An extensive study across 30 popular LLMs using our benchmark ENAMEL shows that LLMs still fall short of generating expert-level efficient code. Benchmarked with our expert-written reference solutions, the strongest commercial LLM GPT-4 has low eff@1=0.454 despite its high pass@1=0.831. Furthermore, using two subsets of our problem set, we show that their deficiency is because LLMs struggle in designing advanced algorithms and are barely aware of implementation optimization.

## 2 EVALUATION FRAMEWORK

Here, we describe our evaluation framework (§2.1), our new efficiency score of a code sample (§2.2), and our new efficiency metric eff@$k$ of an LLM with an unbiased, variance-reduced estimator (§2.3). The main notations used in this paper are summarized in Table 5.

### 2.1 LEVEL-BASED EVALUATION

To achieve a fine-grained evaluation of efficiency, we aim not only to let the most efficient code pass but also to give a continuous score for less efficient code generated by LLMs. A naïve idea is to time each code under large-scale inputs. However, because we have to set a time limit per test case to prevent unacceptably long execution time, if we used only large-scale inputs to evaluate every code, most of the less efficient code would time out, making it impossible to distinguish different efficiencies. For example, for the problem and code samples in Fig. 1, if we used large-scale inputs that allow only the code with $\Theta(\log n)$ iterations to pass, then we would not be able to give different scores for the code with $2^{\Theta(n)}$ recursions and the code with $\Theta(n)$ iterations.

To address this issue, we propose to use multiple levels $1, \ldots, L$ of test cases where each level has a different input scale (i.e., the size of the input). For each problem $i$, all levels share the same time limit $T_i$ while the input scale increases with the level $l$ (i.e., the $L$-th level has the largest input scale). Input scales are carefully designed by a human expert so that algorithms with different efficiencies can pass different numbers of levels. Besides levels $1, \ldots, L$, we use an additional level 0 to filter out wrong code using small strong inputs. For each problem $i$, each level $l = 0, 1, \ldots, L$ has $M_l$ test cases. If the output of the code does not match the expected output in any test case or does not pass level 0, we will not count it into the pass@$k$ metric. If the code passes level 0 but exceeds the time limit in some level $l \geq 1$, we will still count it into the pass@$k$ metric but will skip the remaining levels (i.e., we assume that it will also exceed the time limit for the remaining levels because the input scale increases with the level $l$). Finally, we compute its efficiency score according to §2.2.

**Example.** Fig. 1 illustrates our evaluation framework via HumanEval problem #55 (computing the $n$-th Fibonacci number). Level 0 has $n \leq 10$ so that the naïve recursive algorithm (in $2^{\Theta(n)}$ recursions) can pass; level 1 has $n \leq 30$ so that the dynamic programming algorithm (in $\Theta(n)$

iterations) can pass; level 2 has $n \leq 9000$ so that the matrix exponentiation algorithm (in $\Theta(\log n)$ iterations by repeated squaring) can pass; level 3 has $n \leq 10000$ so that the doubling algorithm (still in $\Theta(\log n)$ iterations yet with a smaller hidden constant in $\Theta$) can pass. These carefully designed levels enable us to differentiate code samples that have different efficiencies.

## 2.2 EFFICIENCY SCORE OF A CODE SAMPLE

A unique challenge in efficiency evaluation is *right-censored* (Bang & Tsiatis, 2000) execution time: when an execution is killed due to exceeding the time limit $T$, we cannot know its actual execution time $t$ and only know that $t \geq T$. For instance, if the generated code contains an infinite loop, the right-censored execution time will be clipped to the time limit while the actual execution time should be infinity. Existing evaluations (Niu et al., 2024; Huang et al., 2024) use the execution time without coping with right censoring and thus overestimate the efficiency.

To appropriately handle right-censored execution time, we aim to propose an efficiency score whose dependence on the execution time vanishes whenever the execution time exceeds the time limit. Thus, for the $j$-th code sample $c_{i,j}$ of problem $i$ and for each level $l$, if the code $c_{i,j}$ is correct, we define the efficiency score $f_{i,j,l}$ by

$$f_{i,j,l} := \frac{(T_i - \max\{t_{i,j,l,m}\}_{m=1}^{M_l})^+}{T_i - \max\{t_{i,l,m}^*\}_{m=1}^{M_l}}, \tag{1}$$

where $t_{i,j,l,m}$ is the execution time of code $c_{i,j}$ for the $m$-th test case in level $l$; $t_{i,l,m}^*$ is the execution time of our reference solution for the $m$-th test case in level $l$; $T_i$ is the time limit of problem $i$; and $(\cdot)^+ := \max\{\cdot, 0\}$. Here, we use $\max\{t_{i,j,l,m}\}_{m=1}^{M_l}$ in $f_{i,j,l}$ to characterize the worst-case efficiency since our expert-written input generators produce various types of test cases that cover the worst cases of various algorithms. Our efficiency score $f_{i,j,l}$ is not affected by right-censored execution time because whenever $\max\{t_{i,j,l,m}\}_{m=1}^{M_l} \geq T_i$, our score $f_{i,j,l}$ will have the same value zero regardless of the exact value of $\max\{t_{i,j,l,m}\}_{m=1}^{M_l}$. Also, we normalize our efficiency score $f_{i,j,l}$ using our reference solution so that the scale of the score does not differ across problems. For the time limit, we use $T_i := \alpha \max\{t_{i,l,m}^*\}_{l,m}$, where $\alpha > 1$ is a hyperparameter. Besides that, to reduce the variance of the execution time caused by hardware performance fluctuations, we repeat each test case $R$ times and estimate the execution time $t_{i,j,l,m}$ via the Hodges–Lehmann estimator (Hodges Jr. & Lehmann, 1963) because of its robustness against outliers as well as its high statistical efficiency.

Finally, since each level has a distinct hardness, we define the efficiency score $e_{i,j}$ of a code sample $c_{i,j}$ of problem $i$ by a weighted average over levels $1, \ldots, L$:

$$e_{i,j} := \begin{cases} \frac{\sum_{l=1}^{L} h_l \cdot f_{i,j,l}}{\sum_{l=1}^{L} h_l}, & \text{if code } c_{i,j} \text{ is correct;} \\ 0, & \text{otherwise.} \end{cases} \tag{2}$$

where hyperparameters $h_l > 0$ represent the hardness of each level $l$.

## 2.3 EFFICIENCY METRIC FOR AN LLM

The pass@$k$ metric (Chen et al., 2021) is the standard metric in correctness evaluation, which means the probability that at least one among $k$ generated code samples is correct. Meanwhile, existing efficiency evaluations (Niu et al., 2024; Huang et al., 2024) use the average execution time as the metric and thus fall short of describing the relationship between code efficiency and sample size $k$.

To overcome this limitation and evaluate the capability of an LLM in generating efficient code w.r.t. the sample size $k$, we aim to generalize the pass@$k$ metric from correctness to our continuous efficiency score. Let $z_i$ denote the prompt of problem $i$; let $c_{i,j} \sim \text{LLM}(z_i)$ denote the generated code samples for problem $i$; let $g_{i,j} \in \{0, 1\}$ denote the correctness of code $c_{i,j}$; and let $\text{pass}_i@k$ denote the pass@$k$ metric w.r.t problem $i$. The original definition of pass@$k$ relies on the Boolean nature of code correctness and thus cannot be directly generalized to our continuous efficiency score.

To address this, we equivalently express $\text{pass}_i@k$ as an expectation:

$$\text{pass}_i@k = \Pr_{c_{i,1},\ldots,c_{i,k} \sim \text{LLM}(z_i)} \left\{ \exists 1 \leq j \leq k : g_{i,j} = 1 \right\} = \Pr_{c_{i,1},\ldots,c_{i,k} \sim \text{LLM}(z_i)} \left\{ \max_{j=1}^{k} g_{i,j} = 1 \right\} \quad (3)$$

$$= \mathbb{E}_{c_{i,1},\ldots,c_{i,k} \sim \text{LLM}(z_i)} \left[ \max_{j=1}^{k} g_{i,j} \right]. \quad (4)$$

This equivalent formula in Eq. (4) no longer relies on the Boolean nature of code correctness and naturally extends to our continuous efficiency score. Hence, we define our efficiency metric $\text{eff}_i@k$ by the expected maximum efficiency score of $k$ independent code samples:

$$\text{eff}_i@k := \mathbb{E}_{c_{i,1},\ldots,c_{i,k} \sim \text{LLM}(z_i)} \left[ \max_{j=1}^{k} e_{i,j} \right], \quad (5)$$

where $e_{i,j}$ denotes the efficiency score of code $c_{i,j}$ defined in §2.2. Our metric $\text{eff}_i@k$ precisely characterizes the relation between code efficiency and sample size $k$ via the maximum over $k$ code samples while the metric in previous works (Niu et al., 2024; Huang et al., 2024) is simply an average over code samples and cannot describe its relation with sample size $k$.

However, estimating $\text{eff}_i@k$ naïvely by generating $k$ code samples and calculating their maximum $e_{i,j}$ can have high variance (Chen et al., 2021). To reduce the variance of $\text{eff}_i@k$ estimation, we employ two advanced variance reduction techniques: (i) bootstrap (Efron, 1979) and (ii) Rao–Blackwellization (Casella & Robert, 1996). Specifically, for $n \geq k$ i.i.d. code samples $c_{i,1},\ldots,c_{i,n} \sim \text{LLM}(z_i)$, the bootstrap estimator is the average of $\max_{j \in J} e_{i,j}$ over multiple random subsets $J \subseteq \{1,\ldots,n\}$ with $|J| = k$, and we obtain our final estimator $\widehat{\text{eff}}_i@k$ by Rao–Blackwellizing the boostrap estimator (i.e., taking expectation over the random subset $J$):

$$\widehat{\text{eff}}_i@k := \mathbb{E}_{\substack{J \subseteq \{1,\ldots,n\} \\ |J|=k}} \left[ \max_{j \in J} e_{i,j} \right] = \sum_{r=k}^{n} \frac{\binom{r-1}{k-1}}{\binom{n}{k}} e_{i,(r)}. \quad (6)$$

where $e_{i,(r)}$ denotes the $r$-th smallest score among $e_{i,1},\ldots,e_{i,n}$, and $\binom{n}{k}$ denotes the binomial coefficient. Furthermore, we show in Theorem 1 that our Rao–Blackwellized bootstrap estimator $\widehat{\text{eff}}_i@k$ is unbiased and does reduce variance.

**Theorem 1.** *Suppose that problem $i$ has time limit $T_i < \infty$ and reference execution times $t^*_{i,l,m} < T_i$. Under the randomness of code generation and execution, for $n \geq k$, we have:*

- *Unbiasedness:*

$$\mathbb{E}_{c_{i,1},\ldots,c_{i,n} \sim \text{LLM}(z_i)} \left[ \sum_{r=k}^{n} \frac{\binom{r-1}{k-1}}{\binom{n}{k}} e_{i,(r)} \right] = \mathbb{E}_{c_{i,1},\ldots,c_{i,k} \sim \text{LLM}(z_i)} \left[ \max_{j=1}^{k} e_{i,j} \right]; \quad (7)$$

- *Variance reduction:*

$$\text{Var}_{c_{i,1},\ldots,c_{i,n} \sim \text{LLM}(z_i)} \left[ \sum_{r=k}^{n} \frac{\binom{r-1}{k-1}}{\binom{n}{k}} e_{i,(r)} \right] \leq \frac{k}{n} \cdot \text{Var}_{c_{i,1},\ldots,c_{i,k} \sim \text{LLM}(z_i)} \left[ \max_{j=1}^{k} e_{i,j} \right]. \quad (8)$$

Proof is in §B. Due to unbiasedness, we will use $\text{eff}_i@k$ and $\widehat{\text{eff}}_i@k$ interchangeably from now on.

As a remark, naïvely computing the coefficients $\binom{r-1}{k-1}/\binom{n}{k}$ in $\widehat{\text{eff}}_i@k$ can result in numerical instability. Instead, we propose a numerically stable implementation of $\widehat{\text{eff}}_i@k$, presented in Algorithm 1.

Finally, we define our efficiency metric $\text{eff}@k$ by averaging $\text{eff}_i@k$ over all problems $i$.

## 3 BENCHMARK DEVELOPMENT

In this section, we detail our methodology for selecting our problemset (§3.1), implementing our efficient reference solutions (§3.2), and curating our strong test case generators (§3.3).

---

**Algorithm 1** Numerically stable $\widehat{\text{eff}}_i@k$

---

**Input:** score list $[e_{i,1}, \ldots, e_{i,n}]$; the target $k$
**Output:** the estimated $\widehat{\text{eff}}_i@k$
1: $\lambda_n \leftarrow \frac{k}{n}$
2: **for** $r \leftarrow n-1, n-2, \ldots, k$ **do**
3: $\quad \lambda_r \leftarrow \lambda_{r+1} \cdot \left(1 - \frac{k-1}{r}\right)$
4: **end for**
5: $[e_{i,(1)}, \ldots, e_{i,(n)}] \leftarrow \text{sort}([e_{i,1}, \ldots, e_{i,n}])$
6: **return** $\sum_{r=k}^{n} \lambda_r e_{i,(r)}$

---

Table 1: A sample of hard problems in our problemset. Our expert-written reference solutions are much more efficient than HumanEval+ canonical solutions. (See Appendix E for code.)

| ID | Problem Description | HumanEval+ Solution | Our Expert Solution |
|----|---------------------|---------------------|---------------------|
| #10 | Find the shortest palindrome that begins with a given string $S$ | $O(|S|^2)$: Enumerate suffixes and check palindromicity | $\Theta(|S|)$: Use Knuth–Morris–Pratt w.r.t. reversed $S$ plus $S$ |
| #36 | Count digit 7's in positive integers $< n$ that are divisible by 11 or 13 | $\Theta(n \log n)$: Enumerate integers $< n$ and count the digits | $\Theta(\log n)$: Design a dynamic programming over digits |
| #40 | Check if a list $l$ has three distinct elements that sum to 0 | $O(|l|^3)$: Enumerate triples in $l$ and check their sums | $O(|l|^2)$: Use a hash set and enumerate pairs in $l$ |
| #109 | Check if a list $a$ can be made non-decreasing using only rotations | $O(|a|^2)$: Enumerate the rotations of $a$ and check | $O(|a|)$: Check if the list $a$ has at most one inversion |
| #154 | Check if any rotation of a string $b$ is a substring of a string $a$ | $O(|b|^2|a|)$: Enumerate rotations and run string matching | $O(|a| + |b|)$: Run the suffix automaton of $a$ w.r.t. $b + b$ |

## 3.1 PROBLEM SELECTION

To achieve a comprehensive evaluation of efficiency, we aim to create a problemset that contains high-quality problems with a broad range of difficulties. Thus, following HumanEval+ (Liu et al., 2023a), we re-use the problems from the HumanEval dataset (Chen et al., 2021) due to their high quality and diverse difficulties. We remark that even seemingly easy problems can become hard if the input scale increases. Although most HumanEval problems *seem* easy, we find that quite a number of them become hard and require advanced algorithms under large-scale inputs. For instance, although the common algorithm for problem #55 (computing the $n$-th Fibonacci number) is dynamic programming with $\Theta(n)$ iterations, a large $n$ requires an advanced doubling algorithm that needs only $\Theta(\log n)$ iterations based on a non-trivial identity of Fibonacci numbers.

Meanwhile, we find that some problems in HumanEval with $\Theta(1)$ time complexity are unsuitable for efficiency evaluation due to the following two reasons. First, their execution time is too short and is thus mainly affected by hardware performance fluctuations, making their execution time un-informative about the true efficiency of the code. Second, since all LLMs do well in these trivial problems, evaluation with these problems hardly differentiates the capabilities of different LLMs. Hence, we exclude these trivial problems and use the remaining 142 problems as our problemset.

Our problemset comprises a wide spectrum of easy to hard problems, thus enabling a comprehensive evaluation of how capable the LLM is in generating efficient code under various difficulties. Table 1 exhibits a sample of hard problems in our problemset.

## 3.2 EFFICIENT REFERENCE SOLUTIONS

An ideal reference of efficiency should be the most efficient code, which often needs advanced algorithms and implementation optimization that can be highly non-trivial even for human programmers. Thus, we employ a human expert to write reference solutions. For each problem, our expert first designs the best algorithm and next optimizes the implementation of the algorithm. Our expert-written reference solutions enable us to evaluate how LLMs compare with human experts in writing efficient code. We introduce our algorithm design stage and implementation optimization stage below.

**Algorithm design.** The goal of algorithm design is to optimize time complexity. It may involve advanced algorithms and non-trivial reformulations, which can be challenging even for human programmers. Thanks to the strong expertise of our human expert, we are able to design the best algorithm as our reference solutions for *all* problems. We remark that we try our best to avoid randomized algorithms whenever an efficient deterministic algorithm exists. Our reference solutions involve many advanced algorithms (such as automata, data structures, and dynamic programming) and a wide range of mathematical knowledge (including number theory, combinatorics, and linear algebra). See Table 1 for a sample of hard problems and our reference solutions.

**Implementation optimization.** Even a single algorithm can have multiple functionally equivalent implementations with different efficiencies. Implementation optimization is to improve code efficiency by exercising best practices and exploiting programming language features, some of which are barely known to non-expert programmers. For example, for problem #98 (counting uppercase vowels at even indices), an efficient Python implementation needs a clever use of the builtin function `str.translate` rather than straightforward counting. To this end, we employ a human expert to find the most efficient implementations as our reference solutions. For each problem, our human expert writes and executes multiple implementations and keeps the most efficient one. Many of our reference solutions are much more efficient than those in HumanEval and HumanEval+ (see Table 2).

### 3.3 STRONG TEST CASE GENERATORS

Previous works either rely on existing HumanEval test cases (Niu et al., 2024), which are known to be not strong enough (Liu et al., 2023a), or use ChatGPT-generated test case generators (Huang et al., 2024), which are mostly random and thus may not differentiate suboptimal algorithms. To address these limitations, we employ a human expert to curate strong test case generators that cover both corner cases to filter out wrong code and worst cases to differentiate suboptimal algorithms. For each problem, our human expert first creates an initial version of the test case generator via ChatGPT and next decides if the problem has corner cases and/or non-random worst cases. If so, then our human expert will strengthen the test case generator by adding such corner cases and/or worst cases. Some corner cases can be non-trivial for non-experts: for example, for problem #31 (deciding if a number is prime), the Fermat primality test is an efficient yet wrong algorithm with only a few non-trivial counterexamples (Carmichael, 1912). As a remark, we only use absolutely valid corner cases and try our best to avoid those whose validity is unclear due to the ambiguity in problem description.

Our expert-written test case generators set a strict and high standard for both correctness and efficiency. For example, 11 canonical solutions in HumanEval and 4 in HumanEval+ are found wrong, and 34 in HumanEval and 27 in HumanEval+ exceed the time limit (see Table 2 for a comparison).

## 4 EVALUATION

We comprehensively evaluate 30 popular LLMs with our ENAMEL benchmark. Due to the space limit, see Appendix C.1 for experimental setting.

Table 2: Comparison with existing benchmarks.

| Name | eff@1 | pass@1 |
|------|-------|--------|
| HumanEval | 0.455 | 0.908 |
| HumanEval+ | 0.513 | 0.972 |
| ENAMEL (ours) | 1.000 | 1.000 |

### 4.1 MAIN RESULTS & ANALYSIS

Table 3 shows pass@$k$ and eff@$k$ of 30 LLMs under our benchmark. Overall, our results suggest that LLMs still fall short of generating expert-level efficient code. Benchmarked with our expert-written reference solutions, even the strongest commercial LLM GPT-4 cannot achieve eff@1>0.5, and most LLMs cannot even reach eff@1>0.3. We also observe that eff@$k$ is *consistently* much lower than pass@$k$ across all LLMs, model sizes, and sample sizes $k$. This stems from the fact that existing research has been primarily focusing on code correctness while overlooking code efficiency, partially due to the lack of a rigorous evaluation framework for code efficiency. Surprisingly, LLMs that are good at generating correct code are not always equally good at generating efficient code. For instance, GPT-4 Turbo has higher eff@1 than GPT-4 although GPT-4 has higher pass@1 than GPT-4 Turbo. A possible reason is that naïve algorithms are easier to be generated correctly but are less

Table 3: Evaluation results under our benchmark. (Greedy: selecting the next token with the highest logit. Sampling: selecting the next token with probability proportional to the softmax of logits.) Existing LLMs fall short of generating expert-level efficient code.

| Model | Greedy | | Sampling | | | | | |
|---|---|---|---|---|---|---|---|---|
| | eff@1 | pass@1 | eff@1 | pass@1 | eff@10 | pass@10 | eff@100 | pass@100 |
| GPT-4 Turbo | 0.470 | 0.796 | — | — | — | — | — | — |
| GPT-4 | 0.454 | 0.831 | — | — | — | — | — | — |
| Llama 3 70B Instruct | 0.421 | 0.746 | 0.438 | 0.747 | 0.526 | 0.836 | 0.575 | 0.880 |
| Llama 3 8B Instruct | 0.344 | 0.592 | 0.345 | 0.564 | 0.500 | 0.770 | 0.595 | 0.874 |
| Mixtral 8x22B Instruct | 0.408 | 0.746 | 0.407 | 0.721 | 0.575 | 0.870 | 0.704 | 0.923 |
| Mixtral 8x7B Instruct | 0.266 | 0.444 | 0.279 | 0.456 | 0.436 | 0.689 | 0.542 | 0.810 |
| Claude 3 Opus | 0.401 | 0.789 | — | — | — | — | — | — |
| Claude 3 Sonnet | 0.345 | 0.662 | 0.365 | 0.677 | 0.498 | 0.814 | 0.594 | 0.887 |
| Claude 3 Haiku | 0.386 | 0.739 | 0.382 | 0.730 | 0.478 | 0.831 | 0.529 | 0.861 |
| Phind Code Llama V2 | 0.394 | 0.683 | 0.372 | 0.638 | 0.584 | 0.862 | 0.723 | 0.935 |
| ChatGPT | 0.364 | 0.683 | 0.374 | 0.673 | 0.557 | 0.847 | 0.690 | 0.937 |
| Code Llama 70B Python | 0.264 | 0.500 | 0.082 | 0.177 | 0.326 | 0.610 | 0.614 | 0.908 |
| Code Llama 34B Python | 0.268 | 0.458 | 0.226 | 0.405 | 0.511 | 0.786 | 0.711 | 0.934 |
| Code Llama 13B Python | 0.216 | 0.408 | 0.204 | 0.372 | 0.487 | 0.732 | 0.714 | 0.899 |
| Code Llama 7B Python | 0.247 | 0.373 | 0.180 | 0.320 | 0.432 | 0.663 | 0.643 | 0.837 |
| StarCoder | 0.195 | 0.352 | 0.134 | 0.236 | 0.355 | 0.557 | 0.542 | 0.787 |
| CodeGen 16B | 0.169 | 0.310 | 0.122 | 0.219 | 0.326 | 0.512 | 0.536 | 0.761 |
| CodeGen 6B | 0.193 | 0.296 | 0.111 | 0.188 | 0.298 | 0.455 | 0.491 | 0.694 |
| CodeGen 2B | 0.153 | 0.254 | 0.098 | 0.168 | 0.264 | 0.389 | 0.421 | 0.602 |
| CodeT5+ 16B | 0.160 | 0.317 | 0.130 | 0.250 | 0.343 | 0.551 | 0.551 | 0.785 |
| Mistral 7B | 0.152 | 0.275 | 0.116 | 0.222 | 0.335 | 0.541 | 0.557 | 0.791 |
| Vicuna 13B | 0.123 | 0.176 | 0.080 | 0.125 | 0.188 | 0.310 | 0.319 | 0.537 |
| Vicuna 7B | 0.061 | 0.099 | 0.054 | 0.081 | 0.149 | 0.231 | 0.283 | 0.423 |
| SantaCoder | 0.100 | 0.141 | 0.088 | 0.126 | 0.204 | 0.298 | 0.349 | 0.470 |
| Incoder 6B | 0.091 | 0.127 | 0.054 | 0.078 | 0.164 | 0.242 | 0.319 | 0.439 |
| Incoder 1B | 0.066 | 0.092 | 0.031 | 0.043 | 0.100 | 0.139 | 0.191 | 0.241 |
| GPT-J | 0.083 | 0.106 | 0.039 | 0.058 | 0.119 | 0.166 | 0.221 | 0.331 |
| GPT-Neo 2B | 0.043 | 0.056 | 0.019 | 0.027 | 0.069 | 0.096 | 0.127 | 0.181 |
| PolyCoder | 0.037 | 0.049 | 0.021 | 0.029 | 0.067 | 0.084 | 0.121 | 0.155 |
| StableLM 7B | 0.020 | 0.021 | 0.007 | 0.010 | 0.039 | 0.048 | 0.097 | 0.123 |

Table 4: Evaluation on two subsets of problems. LLMs struggle in designing advanced algorithms and are largely unaware of implementation optimization. (See Appendix C.2 for the complete table.)

| Model | Algorithm Design Subset | | | | | | Implementation Optimization Subset | | | | | |
|---|---|---|---|---|---|---|---|---|---|---|---|---|
| | eff@1 | pass@1 | eff@10 | pass@10 | eff@100 | pass@100 | eff@1 | pass@1 | eff@10 | pass@10 | eff@100 | pass@100 |
| Llama 3 70B Instruct | 0.246 | 0.660 | 0.306 | 0.749 | 0.359 | 0.750 | 0.404 | 0.791 | 0.497 | 0.869 | 0.551 | 0.920 |
| Llama 3 8B Instruct | 0.201 | 0.518 | 0.303 | 0.724 | 0.367 | 0.849 | 0.313 | 0.582 | 0.468 | 0.806 | 0.571 | 0.906 |
| Mixtral 8x22B Instruct | 0.225 | 0.635 | 0.363 | 0.837 | 0.470 | 0.900 | 0.376 | 0.783 | 0.556 | 0.914 | 0.686 | 0.947 |
| Mixtral 8x7B Instruct | 0.124 | 0.391 | 0.244 | 0.681 | 0.344 | 0.850 | 0.248 | 0.473 | 0.411 | 0.699 | 0.515 | 0.827 |
| Claude 3 Sonnet | 0.184 | 0.577 | 0.328 | 0.804 | 0.450 | 0.950 | 0.358 | 0.723 | 0.475 | 0.846 | 0.548 | 0.893 |
| Claude 3 Haiku | 0.149 | 0.692 | 0.208 | 0.752 | 0.266 | 0.775 | 0.360 | 0.772 | 0.465 | 0.889 | 0.513 | 0.923 |
| Phind Code Llama V2 | 0.185 | 0.554 | 0.353 | 0.789 | 0.401 | 0.849 | 0.351 | 0.712 | 0.567 | 0.901 | 0.732 | 0.968 |
| ChatGPT | 0.120 | 0.488 | 0.304 | 0.799 | 0.483 | 0.950 | 0.337 | 0.715 | 0.508 | 0.864 | 0.633 | 0.949 |
| Code Llama 70B Python | 0.018 | 0.100 | 0.129 | 0.519 | 0.402 | 0.950 | 0.076 | 0.181 | 0.294 | 0.627 | 0.589 | 0.920 |
| Code Llama 34B Python | 0.071 | 0.293 | 0.271 | 0.713 | 0.425 | 0.881 | 0.197 | 0.415 | 0.473 | 0.804 | 0.687 | 0.949 |
| Code Llama 13B Python | 0.058 | 0.212 | 0.276 | 0.665 | 0.478 | 0.844 | 0.176 | 0.405 | 0.476 | 0.784 | 0.715 | 0.928 |
| Code Llama 7B Python | 0.068 | 0.202 | 0.231 | 0.589 | 0.393 | 0.761 | 0.165 | 0.349 | 0.417 | 0.703 | 0.620 | 0.863 |

efficient than advanced algorithms. Besides that, we see that the performance gap between open-source and commercial models are closing in terms of generating efficient code. For example, Phind Code Llama V2 achieves eff@100=0.723, which is even higher than eff@100=0.690 of ChatGPT.

## 4.2 ANALYSIS ON ALGORITHM DESIGN & IMPLEMENTATION OPTIMIZATION

For a more thorough analysis, we further evaluate LLMs on two subsets of our dataset to investigate capabilities in algorithm design and implementation optimization, respectively.

**Algorithm design.** We use a subset consisting of 20 hard problems to evaluate capability in algorithm design. For these problems, the optimal algorithm can have significantly lower time complexity than suboptimal algorithms (see Table 1 for a sample of these problems). Table 4 shows that even when generating 100 samples per problem, the generated code still has low efficiency. For instance, ChatGPT has eff@100=0.483 on this subset, still below 0.5. This suggests that existing LLMs struggle in designing advanced algorithms.

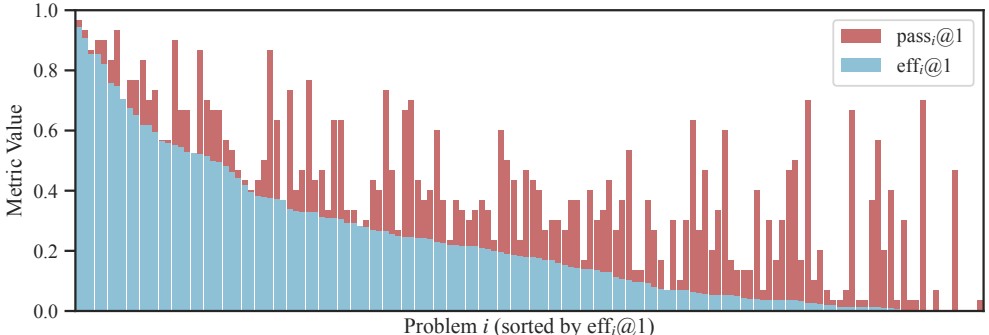

Figure 2: Distribution of problem difficulties (best viewed in color). High $\text{pass}_i@1$ but low $\text{eff}_i@1$ means problem $i$ has a seemingly easy task but a non-trivial efficient algorithm / implementation.

**Implementation optimization.** We use a subset of 75 problems to evaluate the capability in implementation optimization. For these problems, the optimized implementation can have much higher efficiency than naïve implementations. Table 4 shows that the generated code has low efficiency when the sample size is small although the efficiency improves a lot as the sample size increases. For example, Phind Code Llama V2 has good eff@100=0.732 but low eff@1=0.351 over this subset. This suggests that existing LLMs are barely aware of implementation optimization, and the improvement is mainly because random sampling generates multiple equivalent implementations.

### 4.3 DISTRIBUTION OF PROBLEM DIFFICULTIES

To investigate the difficulty distribution of our problems, we plot their $\text{pass}_i@1$ and $\text{eff}_i@1$ (averaged over LLMs under greedy generation) in Fig. 2, where $\text{pass}_i@1$ represents the difficulty of straightforward implementation, and $\text{eff}_i@1$ represents the difficulty of efficient implementation. Fig. 2 demonstrates that our problemset comprises a wide spectrum of easy to hard problems, thus enabling a comprehensive evaluation of capability of LLMs under various difficulties. Notably, some problems $i$ have high $\text{pass}_i@1$ but low $\text{eff}_i@1$ because they have a seemingly easy task with a non-trivial efficient algorithm / implementation. For example, problem #98 (counting uppercase vowels at even indices) has high $\text{pass}_i@1=0.50$ but low $\text{eff}_i@1=0.03$ because an efficient implementation for #98 needs a clever use of builtin functions rather than straightforward counting.

## 5 RELATED WORK

**Code generation.** Code generation (a.k.a. program synthesis) is a long-standing problem in computer science (Simon, 1963). Many classic code generation methods have been proposed over the past few decades (Gulwani et al., 2017), including deductive (Waldinger & Lee, 1969; Manna & Waldinger, 1971; Green, 1981), inductive (Shaw et al., 1975; Gulwani, 2011), and neural-guided approaches (Kalyan et al., 2018; Yu et al., 2023). More recently, many code LLMs have been developed, including Codex (Chen et al., 2021), AlphaCode (Li et al., 2022), CodeGen (Nijkamp et al., 2023), StarCoder (Li et al., 2023), Code Llama (Roziere et al., 2023), CodeT5+ (Wang et al., 2023b), and so on. Some general LLMs such as GPT (OpenAI, 2023), Llama (Meta, 2024), Claude (Anthropic, 2024), Gemini (Google, 2024), and Mixtral (Jiang et al., 2024) also exhibit promising capabilities in code generation.

**Benchmarks for LLM-based code generation.** LLMs have revolutionized machine learning (Wei et al., 2024a;b; 2023; Xu et al., 2024a;b; Chen et al., 2024; Liu et al., 2024a;b;c; 2023b; Qiu et al., 2024b;a; 2023; 2022; Qiu & Tong, 2024; Zeng et al., 2024a;b; Lin et al., 2024a;b; Yoo et al., 2025; 2024; Chan et al., 2024; Wu et al., 2024; He et al., 2024a;b; Wang et al., 2023a; Li et al., 2024b; Qi et al., 2023; Ban et al., 2021). Most of existing benchmarks for LLM-based code generation, including Spider (Yu et al., 2018), HumanEval (Chen et al., 2021), MBPP (Austin et al., 2021), APPS (Hendrycks et al., 2021), MultiPL-E (Cassano et al., 2022), DS-1000 (Lai et al., 2023), HumanEval-X (Zheng et al., 2023), EvalPlus (Liu et al., 2023a), and so on, focus on code correctness. Not until

very recently have a few benchmarks (Nichols et al., 2024; Niu et al., 2024; Huang et al., 2024; Du et al., 2024) been proposed to evaluate code efficiency, and a number of fundamental challenges still remain uncharted and open, which this work aims to address, including how to rigorously handle right-censored execution time, sample size, algorithm/implementation optimization, correctness, and worst-case efficiency. For instance, classic efficiency metrics such as *speedup* (see, e.g., Amdahl, 1967; Touati, 2009) are not designed for right-censored execution time and thus overestimates efficiency when an execution times out.

# 6 CONCLUSION

We have developed a rigorous and high-standard benchmark ENAMEL for evaluating the capability of LLMs in generating *efficient* code, which includes a new metric eff@$k$ (with an unbiased, variance-reduced estimator), expert-written efficient reference solutions for our selected 142 problems, and expert-written strong test case generators. Our extensive evaluation has demonstrated that existing LLMs still fall short of generating expert-level efficient code. We hope LLM developers pay more attention to efficiency of generated code and build more powerful LLMs to reach expert level in the future. Please see Appendix D for limitations and future work.

ACKNOWLEDGMENTS

This work is supported by NSF (2134079), and NIFA (2020-67021-32799), and IBM-Illinois Discovery Accelerator Institute. The content of the information in this document does not necessarily reflect the position or the policy of the Government, and no official endorsement should be inferred. The U.S. Government is authorized to reproduce and distribute reprints for Government purposes notwithstanding any copyright notation here on.

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

Table 5: Nomenclature.

| Symbol | Description |
|---|---|
| $k, n$ | sample sizes |
| $L$ | number of levels |
| $z_i$ | prompt of problem $i$ |
| $c_{i,j}$ | $j$-th code sample for problem $i$ |
| $g_{i,j}$ | correctness of code $c_{i,j}$ |
| $t_{i,j,l,m}$ | execution time of code $c_{i,j}$ for the $m$-th test case at level $l$ |
| $f_{i,j,l}$ | efficiency score of code $c_{i,j}$ at level $l$ |
| $e_{i,j}$ | efficiency score of code $c_{i,j}$ |
| $e_{i,(r)}$ | $r$-th smallest efficiency score among $e_{i,1}, \ldots, e_{i,n}$ |
| $t^*_{i,l,m}$ | reference execution time for the $m$-th test case at level $l$ |
| $T_i$ | time limit of problem $i$ |
| $h_l$ | hardness of level $l$ |
| $M_l$ | number of test cases in level $l$ |
| $\alpha$ | timeout factor |
| $R$ | number of repeats per test case |

# APPENDIX

# A NOMENCLATURE

For reference, the main notations used in this paper are summarized in Table 5.

# B PROOF OF THEOREM 1

In this section, we provide the proofs of unbiasedness and variance reduction, respectively.

## B.1 PROOF OF UNBIASEDNESS

First, recall that every efficiency score $e_{i,j}$ depends only on the corresponding code sample $c_{i,j}$. Since $c_{i,1}, \ldots, c_{i,n}$ are i.i.d., then given any size-$k$ subset $J = \{j_1, \ldots, j_k\} \subseteq \{1, \ldots, n\}$,

$$\mathbb{E}_{c_{i,1},\ldots,c_{i,n}\sim\text{LLM}(z_i)} \left[ \max_{j \in J} e_{i,j} \right] = \mathbb{E}_{c_{i,1},\ldots,c_{i,n}\sim\text{LLM}(z_i)} [\max\{e_{i,j_1}, \ldots, e_{i,j_k}\}] \tag{9}$$

$$= \mathbb{E}_{c_{i,1},\ldots,c_{i,n}\sim\text{LLM}(z_i)} [\max\{e_{i,1}, \ldots, e_{i,k}\}] \tag{10}$$

$$= \mathbb{E}_{c_{i,1},\ldots,c_{i,n}\sim\text{LLM}(z_i)} \left[ \max_{j=1}^{k} e_{i,j} \right] \tag{11}$$

$$= \mathbb{E}_{c_{i,1},\ldots,c_{i,k}\sim\text{LLM}(z_i)} \left[ \max_{j=1}^{k} e_{i,j} \right]. \tag{12}$$

Next, recall that probability measures are finite (and thus $\sigma$-finite). Since efficiency scores $e_{i,j}$ are nonnegative, then by the Fubini–Tonelli theorem and Eq. (12),

$$\mathbb{E}_{c_{i,1},\ldots,c_{i,n}\sim\text{LLM}(z_i)} \left[ \sum_{r=k}^{n} \frac{\binom{r-1}{k-1}}{\binom{n}{k}} e_{i,(r)} \right] = \mathbb{E}_{c_{i,1},\ldots,c_{i,n}\sim\text{LLM}(z_i)} \left[ \mathbb{E}_{\substack{J\subseteq\{1,\ldots,n\}\\|J|=k}} \left[ \max_{j \in J} e_{i,j} \right] \right] \tag{13}$$

$$= \mathbb{E}_{\substack{J\subseteq\{1,\ldots,n\}\\|J|=k}} \left[ \mathbb{E}_{c_{i,1},\ldots,c_{i,n}\sim\text{LLM}(z_i)} \left[ \max_{j \in J} e_{i,j} \right] \right] \tag{14}$$

$$= \mathbb{E}_{\substack{J\subseteq\{1,\ldots,n\}\\|J|=k}} \left[ \mathbb{E}_{c_{i,1},\ldots,c_{i,k}\sim\text{LLM}(z_i)} \left[ \max_{j=1}^{k} e_{i,j} \right] \right] \tag{15}$$

$$= \mathbb{E}_{c_{i,1},\ldots,c_{i,k}\sim\text{LLM}(z_i)} \left[ \max_{j=1}^{k} e_{i,j} \right]. \tag{16}$$

## B.2 PROOF OF VARIANCE REDUCTION

Note that efficiency scores $e_{i,j} \geq 0$ are bounded random variables:

$$e_{i,j} \leq \frac{\sum_{l=1}^{L} h_l \cdot f_{i,j,l}}{\sum_{l=1}^{L} h_l} \leq \max_{l=1}^{L} f_{i,j,l} \tag{17}$$

$$= \max_{l=1}^{L} \frac{(T_i - t_{i,j,l,m})^+}{T_i - \max\{t_{i,l,m}^*\}_{m=1}^{M_l}} \tag{18}$$

$$\leq \max_{l=1}^{L} \frac{T_i - 0}{T_i - \max\{t_{i,l,m}^*\}_{m=1}^{M_l}} < \infty. \tag{19}$$

This implies that

$$\text{Var}_{c_{i,1},\ldots,c_{i,k}\sim\text{LLM}(z_i)} \left[ \max_{j=1}^{k} e_{i,j} \right] < \infty. \tag{20}$$

Furthermore, note that $\widehat{\text{eff}}_i@k$ can be expressed as a U-statistic (Hoeffding, 1948):

$$\sum_{r=k}^{n} \frac{\binom{r-1}{k-1}}{\binom{n}{k}} e_{i,(r)} = \frac{1}{\binom{n}{k}} \sum_{\substack{J\subseteq\{1,\ldots,n\}\\|J|=k}} \max_{j \in J} e_{i,j}. \tag{21}$$

Therefore, by Theorem 5.2 of Hoeffding (1948),

$$\text{Var}_{c_{i,1},\ldots,c_{i,n}\sim\text{LLM}(z_i)} \left[ \sum_{r=k}^{n} \frac{\binom{r-1}{k-1}}{\binom{n}{k}} e_{i,(r)} \right] = \text{Var}_{c_{i,1},\ldots,c_{i,n}\sim\text{LLM}(z_i)} \left[ \frac{1}{\binom{n}{k}} \sum_{\substack{J\subseteq\{1,\ldots,n\}\\|J|=k}} \max_{j \in J} e_{i,j} \right] \tag{22}$$

$$\leq \frac{k}{n} \cdot \text{Var}_{c_{i,1},\ldots,c_{i,k}\sim\text{LLM}(z_i)} \left[ \max_{j=1}^{k} e_{i,j} \right]. \tag{23}$$

Table 6: Complete evaluation results on two subsets of problems.

| Model | Algorithm Design Subset | | | | | | Implementation Optimization Subset | | | | | |
|---|---|---|---|---|---|---|---|---|---|---|---|---|
| | eff@1 | pass@1 | eff@10 | pass@10 | eff@100 | pass@100 | eff@1 | pass@1 | eff@10 | pass@10 | eff@100 | pass@100 |
| Llama 3 70B Instruct | 0.246 | 0.660 | 0.306 | 0.749 | 0.359 | 0.750 | 0.404 | 0.791 | 0.497 | 0.869 | 0.551 | 0.920 |
| Llama 3 8B Instruct | 0.201 | 0.518 | 0.303 | 0.724 | 0.367 | 0.849 | 0.313 | 0.582 | 0.468 | 0.806 | 0.571 | 0.906 |
| Mixtral 8x22B Instruct | 0.225 | 0.635 | 0.363 | 0.837 | 0.470 | 0.900 | 0.376 | 0.783 | 0.556 | 0.914 | 0.686 | 0.947 |
| Mixtral 8x7B Instruct | 0.124 | 0.391 | 0.244 | 0.681 | 0.344 | 0.850 | 0.248 | 0.473 | 0.411 | 0.699 | 0.515 | 0.827 |
| Claude 3 Sonnet | 0.184 | 0.577 | 0.328 | 0.804 | 0.450 | 0.950 | 0.358 | 0.723 | 0.475 | 0.846 | 0.548 | 0.893 |
| Claude 3 Haiku | 0.149 | 0.692 | 0.208 | 0.752 | 0.266 | 0.775 | 0.360 | 0.772 | 0.465 | 0.889 | 0.513 | 0.923 |
| Phind Code Llama V2 | 0.185 | 0.554 | 0.353 | 0.789 | 0.401 | 0.849 | 0.351 | 0.712 | 0.567 | 0.901 | 0.732 | 0.968 |
| ChatGPT | 0.120 | 0.488 | 0.304 | 0.799 | 0.483 | 0.950 | 0.337 | 0.715 | 0.508 | 0.864 | 0.633 | 0.949 |
| Code Llama 70B Python | 0.018 | 0.100 | 0.129 | 0.519 | 0.402 | 0.950 | 0.076 | 0.181 | 0.294 | 0.627 | 0.589 | 0.920 |
| Code Llama 34B Python | 0.071 | 0.293 | 0.271 | 0.713 | 0.425 | 0.881 | 0.197 | 0.415 | 0.473 | 0.804 | 0.687 | 0.949 |
| Code Llama 13B Python | 0.058 | 0.212 | 0.276 | 0.665 | 0.478 | 0.844 | 0.176 | 0.405 | 0.476 | 0.784 | 0.715 | 0.928 |
| Code Llama 7B Python | 0.068 | 0.202 | 0.231 | 0.589 | 0.393 | 0.761 | 0.165 | 0.349 | 0.417 | 0.703 | 0.620 | 0.863 |
| StarCoder | 0.047 | 0.161 | 0.156 | 0.485 | 0.257 | 0.709 | 0.112 | 0.247 | 0.332 | 0.598 | 0.514 | 0.802 |
| CodeGen 16B | 0.031 | 0.133 | 0.146 | 0.451 | 0.292 | 0.684 | 0.099 | 0.220 | 0.303 | 0.541 | 0.531 | 0.801 |
| CodeGen 6B | 0.023 | 0.091 | 0.106 | 0.372 | 0.235 | 0.612 | 0.090 | 0.188 | 0.285 | 0.478 | 0.483 | 0.731 |
| CodeGen 2B | 0.036 | 0.131 | 0.121 | 0.387 | 0.193 | 0.644 | 0.081 | 0.160 | 0.256 | 0.400 | 0.410 | 0.610 |
| CodeT5+ 16B | 0.043 | 0.192 | 0.173 | 0.509 | 0.321 | 0.673 | 0.106 | 0.257 | 0.313 | 0.581 | 0.536 | 0.845 |
| Mistral 7B | 0.030 | 0.152 | 0.157 | 0.516 | 0.319 | 0.737 | 0.100 | 0.227 | 0.327 | 0.574 | 0.565 | 0.821 |
| Vicuna 13B | 0.008 | 0.072 | 0.033 | 0.269 | 0.076 | 0.449 | 0.056 | 0.096 | 0.168 | 0.288 | 0.316 | 0.569 |
| Vicuna 7B | 0.019 | 0.071 | 0.083 | 0.241 | 0.113 | 0.300 | 0.031 | 0.061 | 0.121 | 0.215 | 0.260 | 0.439 |
| SantaCoder | 0.037 | 0.102 | 0.101 | 0.316 | 0.203 | 0.493 | 0.069 | 0.114 | 0.203 | 0.308 | 0.357 | 0.488 |
| Incoder 6B | 0.010 | 0.050 | 0.062 | 0.203 | 0.112 | 0.325 | 0.037 | 0.062 | 0.152 | 0.252 | 0.320 | 0.477 |
| Incoder 1B | 0.003 | 0.023 | 0.021 | 0.110 | 0.071 | 0.200 | 0.018 | 0.030 | 0.080 | 0.129 | 0.172 | 0.232 |
| GPT-J | 0.021 | 0.051 | 0.063 | 0.146 | 0.081 | 0.243 | 0.025 | 0.043 | 0.110 | 0.167 | 0.221 | 0.354 |
| GPT-Neo 2B | 0.003 | 0.019 | 0.015 | 0.098 | 0.032 | 0.172 | 0.007 | 0.014 | 0.050 | 0.084 | 0.113 | 0.184 |
| PolyCoder | 0.002 | 0.010 | 0.018 | 0.070 | 0.050 | 0.163 | 0.004 | 0.007 | 0.034 | 0.051 | 0.092 | 0.122 |
| StableLM 7B | 0.001 | 0.005 | 0.010 | 0.039 | 0.033 | 0.099 | 0.002 | 0.003 | 0.016 | 0.025 | 0.074 | 0.099 |

# C  EVALUATION (CONTINUED)

## C.1  EXPERIMENTAL SETTING

**Code generation.** For models that are included in Liu et al. (2023a), we re-use their generated code samples. For other open-source models, we use temperature 0.8 and top_p 0.95 for sampling on a server with 8 NVIDIA A100 80GB GPUs. For Claude 3 models, we use the API provided by Anthropic with temperature 0.8 for sampling. Due to financial and computational constraints, for relatively smaller models, we generate 200 code samples per problem under sampling; for larger models, we generate 100 code samples per problem under sampling; for largest commercial models, we only use greedy decoding. In our experiments, Claude 3 Opus refers to `claude-3-opus-20240229`; Claude 3 Sonnet refers to `claude-3-sonnet-20240229`; Claude 3 Haiku refers to `claude-3-haiku-20240307`; GPT-4 Turbo refers to `gpt-4-1106-preview`; GPT-4 refers to `gpt-4-0613`.

**Code evaluation.** We use $\alpha = 2$, $R = 6$, $h_1 = h_2 = 3$, $h_3 = 4$, $M_0 = 8$, $M_1 = M_2 = M_3 = 4$. To minimize server workload fluctuations, we run evaluation on virtualized cloud servers hosted by Google Cloud (Ubuntu 20.04.6 LTS; Intel Xeon CPU @ 2.20GHz; Python 3.10.12). We use the reference time on the slowest test case for each problem to further calibrate the execution time of generated code.

**Use of existing assets.** Our benchmark partially uses problems from HumanEval (Chen et al., 2021; MIT License) and prompts from HumanEval+ (Liu et al., 2023a; Apache License). Some reference solutions are modified based on the canonical solutions in HumanEval and HumanEval+.

## C.2  ANALYSIS ON ALGORITHM DESIGN & IMPLEMENTATION OPTIMIZATION (CONTINUED)

The complete version of Table 4 is shown in Table 6. We can see that observations for Table 6 are similar with those for Table 4.

## C.3  COMPARISON OF EFFICIENCY METRICS

To demonstrate that our proposed eff@$k$ metric can rigorously handle right-censored execution times, we empirically compare our eff@$k$ with a classic metric called *speedup* (Amdahl, 1967). The speedup metric is originally defined as the execution time $t^*_{i,l,m}$ of the reference solution divided by the true execution time $t_{i,j,l,m}$ of the generated code. Nonetheless, since generated code can exceed the time limit $T_i$ in our evaluation, the actual definition of speedup is $\frac{t^*_{i,l,m}}{\min\{t_{i,j,l,m}, T_i\}}$

Table 7: Comparison of our proposed efficiency metric and the classic speedup metric. Different rankings are marked in **bold** font. Under the speedup metric, Mixtral 8x22B Instruct and Llama 3 70B Instruct even *seems* to outperform GPT-4.

| Rank | eff@1 (ours) | speedup |
|:---:|---|---|
| 1 | GPT-4 Turbo | GPT-4 Turbo |
| 2 | **GPT-4** | **Mixtral 8x22B Instruct** |
| 3 | Llama 3 70B Instruct | Llama 3 70B Instruct |
| 4 | **Mixtral 8x22B Instruct** | **GPT-4** |
| 5 | Claude 3 Opus | Claude 3 Opus |
| 6 | Phind Code Llama V2 | Phind Code Llama V2 |
| 7 | **Claude 3 Haiku** | **ChatGPT** |
| 8 | **ChatGPT** | **Claude 3 Haiku** |
| 9 | Claude 3 Sonnet | Claude 3 Sonnet |
| 10 | Llama 3 8B Instruct | Llama 3 8B Instruct |
| 11 | **Code Llama 34B Python** | **Mixtral 8x7B Instruct** |
| 12 | **Mixtral 8x7B Instruct** | **Code Llama 34B Python** |

Table 8: Comparison between the random test generator and our expert-written test case generator on problem #31. Better results are marked in **bold** font. Random test cases cannot assess true correctness or true efficiency while our test case generator can.

| Generator | Naïve | Fermat |
|---|:---:|:---:|
| Random | 0.91 | 1.25 |
| Expert (ours) | **0.17** | **0.00** |

instead, which overestimates efficiency when $t_{i,j,l,m} > T_i$. We average the speedup score over all test cases in each level, and we use the same hardnesses $h_1, h_2, h_3$ to weigh the levels.

Table 7 shows rankings of LLMs with greedy decoding under our eff@1 metric and the speedup metric, respectively. We can see that eff@1 and speedup give very different rankings, especially for top-performing LLMs. In particular, under the speedup metric, Mixtral 8x22B Instruct and Llama 3 70B Instruct even *seems* to outperform GPT-4. The unreasonable ranking by the speedup metric is because the speedup metric overestimates efficiency in the presence of right-censored execution time (i.e., when the program exceeds the time limit), as we discussed above. Therefore, it is necessary to propose our eff@$k$ metric to more rigorously handle right-censored execution time.

## C.4 COMPARISON WITH RANDOM TEST CASES

To further demonstrate the strength of our expert-written test case generators, we provide a case study comparing our strong generator and the random test case generator for the problem #31 (deciding if a number $n$ is prime). We investigate the following two solutions: (i) Naïve: the $O(n)$-time factorization algorithm, which is correct but inefficient; (ii) Fermat: the Fermat primality test (Carmichael, 1912), which is efficient but wrong. We compare the eff@1 metrics of these two solutions under the random generator and our test case generator, respectively. Results are shown in Table 8. We can see that random test cases cannot assess true correctness or true efficiency while our test case generator can. This demonstrates the strength of our expert-written test case generators.

## C.5 COMPARISON WITH OTHER BENCHMARKS

To further demonstrate the difficulty of our problems, we provide a comparison of evaluation results between EffiBench, Mercury, and our benchamrk ENAMEL. Since Mercury did not evaluate GPT-4 or Code Llama 70B Python, we compare the reported evaluation results of Code Llama 34B Python. The results are shown in Table 9. From the table, we can see that EffiBench's and Mercury's LeetCode problems are less challenging to LLMs than our benchmark. This is because LLMs (i)

Table 9: Comparison with other efficiency benchmarks. The most challenging benchmark is marked in **bold** font. Our benchmark ENAMEL is more challenging than EffiBench and Mercury.

| Benchmark | Metric | Result |
|---|---|---|
| EffiBench | 1/NET | 0.336 |
| Mercury | Beyond | 0.424 |
| **ENAMEL (ours)** | eff@1 | **0.268** |

Table 10: Analysis of timeout factor $\alpha$ and hardnesses $h_1, h_2, h_3$ on GPT-4 Turbo.

(a) Timeout factor $\alpha$.

| $\alpha$ | 1.5 | 2.0 | 2.5 | 3.0 | 3.5 |
|---|---|---|---|---|---|
| eff@1 | 0.421 | 0.470 | 0.502 | 0.525 | 0.541 |

(b) Level-1 hardness $h_1$.

| $h_1$ | 1 | 2 | 3 | 4 | 5 |
|---|---|---|---|---|---|
| eff@1 | 0.428 | 0.451 | 0.470 | 0.486 | 0.498 |

(c) Level-2 hardness $h_2$.

| $h_2$ | 1 | 2 | 3 | 4 | 5 |
|---|---|---|---|---|---|
| eff@1 | 0.474 | 0.472 | 0.470 | 0.469 | 0.467 |

(d) Level-3 hardness $h_3$.

| $h_3$ | 1 | 2 | 3 | 4 | 5 |
|---|---|---|---|---|---|
| eff@1 | 0.520 | 0.499 | 0.483 | 0.470 | 0.460 |

have seen the public solutions on LeetCode (editorials and community solutions) but (ii) have never seen our expert-written efficient solutions.

## C.6 ANALYSIS OF HYPERPARAMETERS

Our benchmark has timeout factor $\alpha$ and hardnesses $h_1, h_2, h_3$ as hyperparameters. Regarding the timeout factor $\alpha$, it represents the tolerance to execution timeout because the execution time limit is proportional to $\alpha$. Thus, if one wants to tolerate less efficient code, then they can use a larger $\alpha$. Regarding hardnesses $h_1, h_2, h_3$, it represents how we weigh each level. Thus, if one wants to focus more on easier levels, they should use a larger $h_1$; if one wants to focus more on harder levels, they should use a larger $h_3$. We encourage users to stick to our default hyperparameters $\alpha = 2$, $h_1 = 3$, $h_2 = 3$, $h_3 = 4$ to ensure consistency across different test cases and different LLMs. We used these default hyperparameters throughout this work.

To further illustrate how eff@k is influenced by $\alpha$ and $h_1, h_2, h_3$, we report the eff@1 of GPT-4 Turbo with greedy decoding under different $\alpha$, $h_1$, $h_2$, and $h_3$. Results are shown in Table 10. We can see that eff@1 increases as $\alpha$ increases (because alpha represents the tolerance to less efficient code), that eff@1 increases as $h_1$ increases (because we weigh more on an easier level), and that eff@1 decreases as $h_2$ or $h_3$ increases (because we weigh more on a harder levels). These empirical results are consistent with the aforementioned analysis. We hope these empirical results will help users decide hyperparameters based on their preferences about the tolerance to execution time and weights across different levels.

## C.7 ANALYSIS OF RAO–BLACKWELLIZATION

To demonstrate that Rao–Blackwellization does reduce the variance of the eff@$k$ estimator, we empirically compute the standard deviation of the vanilla eff@$k$ estimator Eq. (5) and our Rao–

Table 11: Comparison of the standard deviations of the vanilla eff@$k$ estimator and our Rao–Blackwellized eff@$k$ estimator. Better results are marked in **bold** font. Our Rao–Blackwellized estimator achieves significantly lower standard deviation than the vanilla estimator.

| Estimator | $k = 1$ | $k = 10$ |
|---|---|---|
| Vanilla | 0.20 | 0.25 |
| Rao–Blackwellized | **0.02** | **0.08** |

Table 12: Analysis of whether encouraging code efficiency by prompting can improve the efficiency of generated code or not. Results show that encouraging LLMs to generate "the most efficient algorithm" can barely enhance the efficiency of generated code.

(a) Llama 3 70B Instruct.

| Prompt | eff@1 | pass@1 |
|---|---|---|
| Basic | 0.421 | 0.746 |
| Encouraging | 0.418 | 0.746 |

(b) Mixtral 8x22B Instruct.

| Prompt | eff@1 | pass@1 |
|---|---|---|
| Basic | 0.408 | 0.746 |
| Encouraging | 0.426 | 0.732 |

Blackwellized eff@$k$ estimator using the Llama 3 70B Instruct model. We randomly sample 1000 subsets of size $k$ from the 100 generated samples to estimate the standard deviation of the vanilla eff@$k$ estimator. Results are shown in Table 11. We can see that the Rao–Blackwellized estimator achieves much lower standard deviation than the vanilla estimator. Therefore, our Rao–Blackwellized estimator empirically ensures a more accurate estimation of eff@$k$.

### C.8 EVALUATION UNDER PROMPTING ENGINEERING

To investigate how prompt engineering affects evaluation results, we provide the following two experiments on prompt engineering.

**Experiment I: encouraging efficiency.** We use the two strongest open-source LLMs Llama 3 70B Instruct and Mixtral 8x22B Instruct (according to Table 3). We ask the LLM to generate "the most efficient algorithm" and use greedy decoding to obtain outputs. The eff@1 results are presented in Table 12. The results show that this prompt engineering cannot enhance code efficiency much, even for such strong LLMs. The eff@1 of Mixtral 8x22B Instruct increases a little bit but is still far from the expert level. Interestingly, the eff@1 of Llama 3 70B Instruct even drops a little bit while the pass@1 does not change. This suggests that even such strong LLMs lack a good understanding of code efficiency.

**Experiment II: adaptive prompting.** We believe that prompting alone does not fully address the code efficiency issue because this is essentially a fundamental capability limitation of existing LLMs—efficient algorithms can be so non-trivial that existing LLMs do not understand them well. To demonstrate this, we further conduct a case study for a hard problem #36 under the following two settings: (i) Self-Refine prompting (Madaan et al., 2024) and (ii) revealing the efficient algorithm in the prompt. (The efficient algorithm for #36 is a dynamic programming over digits in $O(\log n)$ time; see Appendix E.2).

For Self-Refine, following Niu et al. (2024), we ask the LLM to "give a potential strategy to improve the efficiency of the code" and finally to "give the optimized version of the same code with the strategy mentioned above." Outputs before and after Self-Refine are shown in Table 13. Before Self-Refine, both Llama 3 70B Instruct and Mixtral 8x22B Instruct generate the brute-force algorithm that runs in $O(n)$ time. After Self-Refine, both LLMs apply a small improvement to the generated code, but the time complexity is still $O(n)$, significantly slower than the efficient algorithm that runs in $O(\log n)$ time. This suggests that both LLMs are unable to come up with the efficient algorithm on their own.

To further investigate whether LLMs can generate the dynamic programming algorithm here, we explicitly reveal the efficient algorithm in the prompt. Specifically, we use "Complete the following function with a dynamic programming over digits" for both LLMs. Raw outputs are shown in

Table 13: Results of Self-Refine prompting (Madaan et al., 2024) for problem #36. Both LLMs are unable to come up with the efficient algorithm on their own.

(a) Initial code by Llama 3 70B Instruct.

```
1    count = 0
2    for i in range(n):
3        if i % 11 == 0 or i % 13 == 0:
4            count += str(i).count('7')
5    return count
```

(b) Self-Refined code by Llama 3 70B Instruct.

```
1    import math
2    return sum(str(i).count('7') for i in range(n) if math
     .gcd(i, 11 * 13) in (11, 13))
```

(c) Initial code by Mixtral 8x22B Instruct.

```
1    count = 0
2    for i in range(11, n, 1):
3        if i % 11 == 0 or i % 13 == 0:
4            count += str(i).count('7')
5    return count
```

(d) Self-Refined code by Mixtral 8x22B Instruct.

```
1    count = 0
2    for i in range(11, n, 11):
3        count += str(i).count('7')
4    for i in range(13, n, 13):
5        if i % 11 != 0:  # to avoid double counting
     numbers that are divisible by both 11 and 13
6            count += str(i).count('7')
7    return count
```

Tables 14 & 15. Llama 3 70B Instruct fails to use dynamic programming; Mixtral 8x22B Instruct uses a trivial dynamic programming that is not over digits. We can see that both LLMs still generate inefficient algorithms that run in $O(n)$ time even though we have revealed the efficient $O(\log n)$ algorithm to them. This suggests that they do not have a good understanding of this non-trivial dynamic programming.

From the two case studies above, we can see that, no matter whether we use (i) or (ii), the generated codes are still inefficient algorithms in $O(n)$ time, which are far from the efficient $O(\log n)$ time complexity. Therefore, the code inefficiency is essentially a capability limitation of existing LLMs, and therefore we do not expect it to be easily addressed merely via prompting techniques. This is also our core motivation for developing this benchmark to compare LLMs with human experts. While how to address this capability limitation is beyond the scope of this work, we hope that our work will motivate LLM developers pay more attention to this limitation and try to address it in future work.

## D CONCLUDING REMARKS

### D.1 SCALABILITY OF BENCHMARK DEVELOPMENT

This work employs human expertise to develop high-quality reference solutions and test case generators. We believe that human expert is necessary to develop a high-standard and rigorous benchmark. For example, as shown in Table 2, compared with our expert solutions, HumanEval canonical solutions achieve only eff@1=0.455, and HumanEval+ canonical solutions achieve only eff@1=0.513. This shows that their canonical solutions are far less efficient than our expert-written reference solutions. In fact, we have excluded a few options when designing the benchmark development methodology:

Table 14: Raw output of Llama 3 70B Instruct when we reveal the efficient algorithm in the prompt.

```
Here is the Python function that uses dynamic programming over digits to
    solve the problem:
```python
def fizz_buzz(n: int):
    count = 0
    for i in range(n):
        if i % 11 == 0 or i % 13 == 0:
            count += str(i).count('7')
    return count
```
Here's an explanation of the code:

1. We initialize a variable `count` to 0, which will store the total
    count of digit 7 appearances.
2. We iterate over the range of integers from 0 to `n-1` (since we want
    to consider integers less than `n`).
3. For each integer `i` in the range, we check if it is divisible by 11
    or 13 using the modulo operator (`%`). If it is, we proceed to the
    next step.
4. We convert the integer `i` to a string using the `str()` function, and
     then use the `count()` method to count the number of occurrences of
    the digit '7' in the string.
5. We add the count of digit 7 appearances in the current integer to the
    overall `count` variable.
6. Finally, we return the total `count` at the end of the function.

This function has a time complexity of O(n), where n is the input integer
    , since we iterate over the range of integers from 0 to `n-1`. The
    space complexity is O(1) since we only use a single variable to store
     the count.

You can test the function with the provided examples to verify that it
    produces the correct output:
```python
print(fizz_buzz(50))  # 0
print(fizz_buzz(78))  # 2
print(fizz_buzz(79))  # 3
```
```

- We did not use problems or solutions from online judges (like LeetCode or Codeforces) because their public solutions are already in LLMs' pretraining corpuses. For example, DeepMind's AlphaCode (Li et al., 2022) has been trained on many online judges including Codeforces, CodeChef, HackerEarth, AtCoder, and Aizu. If we evaluate LLMs on these public online judges, then the evaluation results may fail to reflect the LLMs' true capabilities due to test set leakage.

- We did not crowd-source the benchmark because otherwise it would be hard to guarantee the quality of the benchmark. For example, MBPP (Austin et al., 2021) is a popular crowd-sourced benchmark, but it is known to be easier than HumanEval (Roziere et al., 2023).

- We did not use LLM-generated reference solutions because LLM-generated code are still far from expert-level efficiency, as demonstrated in Table 3.

Despite the size of the benchmark, our 142 problems has already revealed the limited capability of all the 30 LLMs in generating efficient code. In particular, our benchmark shows that even the strongest LLM GPT-4 Turbo is still far from generating expert-level efficient code (with eff@1 below 0.5). We hope our findings and benchmark will help LLM developers to realize this critical issue and further inspire them to develop stronger LLM code generators. The effectiveness of our benchmark is because our human expert has carefully verified the comprehensiveness of the problemset:

Table 15: Raw output of Mixtral 8x22B Instruct when we reveal the efficient algorithm in the prompt.

```
Here is a Python function that uses dynamic programming to solve the
    problem:

```python
def fizz_buzz(n: int):
    """Return the number of times the digit 7 appears in integers less
    than n which are divisible by 11 or 13.
    """
    # Initialize a list to store the count of 7s in each number
    dp = [0] * n

    # Base case: the number of 7s in 0 is 0
    dp[0] = 0

    # Iterate over the range of numbers
    for i in range(1, n):
        # If the current number is divisible by 11 or 13
        if i % 11 == 0 or i % 13 == 0:
            # If the last digit of the current number is 7
            if i % 10 == 7:
                # Add 1 to the count of 7s in the current number
                dp[i] = dp[i - 1] + 1
            else:
                # Otherwise, the count of 7s in the current number is the
    same as the previous number
                dp[i] = dp[i - 1]
        else:
            # If the current number is not divisible by 11 or 13, the
    count of 7s is the same as the previous number
            dp[i] = dp[i - 1]

    # Return the count of 7s in numbers less than n
    return dp[n - 1]
```

This function works by iterating over the range of numbers from 1 to n-1
    and checking if each number is divisible by 11 or 13. If it is, the
    function checks if the last digit of the number is 7. If it is, the
    function adds 1 to the count of 7s in the current number. Otherwise,
    the count of 7s in the current number is the same as the previous
    number. The function then returns the count of 7s in numbers less
    than n.

Here are the test cases:

```python
print(fizz_buzz(50))  # Output: 0
print(fizz_buzz(78))  # Output: 2
print(fizz_buzz(79))  # Output: 3
```
```

- As shown in Figure 2, our benchmark problems have diverse difficulties. For example, 75 seemingly easy problems require non-trivial implementation optimization, and 20 hard problems require advanced algorithms.

- As discussed in Section 3.2, our problemset covers a wide range of algorithmic knowledge (including data structures, dynamic programming, and automata) and a wide range of mathematical knowledge (including linear algebra, combinatorics, and number theory).

That said, we still believe that addressing scalability of benchmark development is an important future direction. A possible solution is to collaborate with private programming competitions whose solutions are not publicly available.

### D.2 OTHER LIMITATIONS & FUTURE WORK

The following are other limitations of this work that we also wish to be addressed in future work:

- This work considers standalone programming problems. Meanwhile, real-world software development typically involves complex dependencies among files. Thus, it is worth studying how to generalize our methodology to more complex code generation datasets such as DevBench (Li et al., 2024a).

- Although we have used the known best algorithms as our reference solutions, it is hard to theoretically guarantee their optimality. Thus, the efficiency score can be greater than 1 if the benchmarked code is more efficient than our reference solution. Addressing this issue in future work will provide a solid ground for efficiency evaluation.

- This work focuses on benchmarking code efficiency without more advanced prompting techniques. Future work can explore how to design prompts to improve the efficiency of LLM-generated code. A possible solution is to guide the LLM to analyze the time complexity in the chain of thought (Wei et al., 2022) when generating the code.

- While our current benchmark focuses on evaluating time efficiency, we believe that evaluating the space efficiency would be a very interesting and important future research direction. For example, EffiBench (Huang et al., 2024) is a time–space joint evaluation benchmark for LLM-generated code. A potential challenge is how to evaluate the time–space trade-off. Since many time-efficient algorithms trade space for time (e.g., dynamic programming), a space-optimal algorithm may be less time-efficient, and vice versa. Hence, different reference solutions might be needed for time evaluation and space evaluation, respectively.

- How to developing an automatic method to measure the time complexity will also be a very interesting future direction. Although this might require an independent new study, there are two possible approaches (although both of them have limitations). (i) Time complexity prediction: A possible approach is to train an LLM to predict the time complexity of a given code sample. However, existing time complexity analyzers (such as LeetCode's analyzer) are known to be inaccurate. We believe that time complexity prediction is in general difficult for LLMs (and even diffcult for non-expert humans). For example, the Splay tree (Sleator & Tarjan, 1985) seems to have $O(n)$ time complexity per operation, but a sophisticated analysis by the authors shows that it actually has $O(\log n)$ time complexity per operation. (ii) Fitting a time curve: Another possible approach is to fit a curve of the running time v.s. the input size to help decide the time complexity. However, we believe that this is in general difficult because it is practically infeasible to distinguish a high-degree polynomial from an exponential function. For example, the Agrawal–Kayal–Saxena primality test (Agrawal et al., 2004) runs in $\tilde{O}((\log n)^{12})$ time, so the curve of its running time v.s. $n$ looks extremely like an exponential function for most practical $n$.

## E CODE OF EXAMPLE PROBLEMS IN TABLE 1

### E.1 HUMANEVAL PROBLEM #10

**Problem description:** Find the shortest palindrome that begins with a given `string` ($S$).

**HumanEval+ canonical solution:** Enumerate suffixes and check palindromicity. The time complexity is $O(|S|^2)$.

```
1  def is_palindrome(string: str) -> bool:
2    return string == string[::-1]
3  if is_palindrome(string):
4    return string
5  for i in range(len(string)):
6    if is_palindrome(string[i:]):
7      return string + string[i-1::-1]
```

**Our expert-written solution:** Note that the answer is the concatenation of the *border* of reversed $S$ plus $S$ and reversed $S$, so we can use the Knuth–Morris–Pratt algorithm to compute the border of reversed $S$ plus $S$. The time complexity is $\Theta(|S|)$.

```
1  if not string:
2     return string
3  reversed_s = string[:: -1]
4  pattern = reversed_s + '\x00' + string
5  m = len(pattern)
6  # Knuth--Morris--Pratt
7  fail = [0] * (m + 1)
8  j = 0
9  for i in range(1, m):
10    c = pattern[i]
11    while j > 0 and pattern[j] != c:
12      j = fail[j]
13    if j > 0 or pattern[0] == c:
14      j += 1
15    fail[i + 1] = j
16 return string[: len(string) - fail[-1]] + reversed_s
```

## E.2  HUMANEVAL PROBLEM #36

**Problem description:** Count digit 7's in positive integers $< n$ that are divisible by 11 or 13.

**HumanEval+ canonical solution:** Enumerate integers $< n$ and count the digits. Since the length of the integer $n$ is $\Theta(\log n)$, the time complexity is $\Theta(n \log n)$.

```
1  cnt = 0
2  for i in range(n):
3    if i % 11 == 0 or i % 13 == 0:
4      cnt += len(list(filter(lambda c: c == "7", str(i))))
5  return cnt
```

**Our expert-written solution:** Design a dynamic programming over digits. Since 10, 11, and 13 are constants, the time complexity is $\Theta(\log n)$, proportional to the length of the integer $n$.

```
1  a = []
2  while n > 0:
3    n, u = divmod(n, 10)
4    a.append(u)
5  m = len(a)
6  b = [[1, 1]] # [10 ** i % 11, 10 ** i % 13]
7  for i in range(m - 1):
8    b.append([(b[i][0] * 10) % 11, (b[i][1] * 10) % 13])
9  f = [[[[0, 0] for w in range(10)] for v in range(13)] for u in range
      (11)] for i in range(m)] # [i-th][mod 11, mod 13][digit]: [number of
      valid numbers, number of 7's in valid numbers]
10 for u in range(10):
11   f[0][u][u] = [[int(w >= u), int(u == 7 and w >= 7)] for w in range
      (10)]
12 for i in range(1, m):
13   for u in range(11):
14     for v in range(13):
15       f0 = f[i - 1][u][v][9]
16       for w in range(10):
17         f1 = f[i][(u + b[i][0] * w) % 11][(v + b[i][1] * w) % 13][w]
18         f1[0] += f0[0]
19         f1[1] += f0[1] + f0[0] * int(w == 7)
20     for u in range(11):
21       for v in range(13):
22         f1 = f[i][u][v]
23         for w in range(1, 10):
24           f1[w][0] += f1[w - 1][0]
25           f1[w][1] += f1[w - 1][1]
```

```
26  e = [[0, 0, 0] for i in range(m)]
27  for i in range(m - 1, 0, -1):
28    e[i - 1] = [(e[i][0] + b[i][0] * a[i]) % 11, (e[i][1] + b[i][1] * a[i
      ]) % 13, e[i][2] + int(a[i] == 7)]
29  ans = 0
30  for i in range(m):
31    if a[i]:
32      w = a[i] - 1
33      u = (-e[i][0]) % 11
34      for v in range(13):
35        f1 = f[i][u][v][w]
36        ans += f1[1] + f1[0] * e[i][2]
37      u0 = u
38      v = (-e[i][1]) % 13
39      for u in range(11):
40        if u != u0:
41          f1 = f[i][u][v][w]
42          ans += f1[1] + f1[0] * e[i][2]
43  return ans
```

### E.3   HUMANEVAL PROBLEM #40

**Problem description:** Check if a list $l$ has three distinct elements that sum to 0.

**HumanEval+ canonical solution:** Enumerate triples in $l$ and check their sums. The time complexity is $O(|l|^3)$.

```
1  for i in range(len(l)):
2    for j in range(len(l)):
3      for k in range(len(l)):
4        if i != j and i != k and j != k and l[i] + l[j] + l[k] == 0:
5          return True
6  return False
```

**Our expert-written solution:** Note that $l_i + l_j + l_k = 0$ is equivalent to $l_k = -l_i - l_j$, so we can enumerate $l_i, l_j$, store $-l_i - l_j$ in a hash set, and check whether $l_k$ is in the hash set. The time complexity is $O(|l|^2)$.

```
1  n = len(l)
2  if n < 3:
3    return False
4  for i, x in enumerate(l[: n - 2]):
5    buf = set()
6    for y in l[i + 1 :]:
7      if y in buf:
8        return True
9      buf.add(-x - y)
10 return False
```

### E.4   HUMANEVAL PROBLEM #109

**Problem description:** Check if a list `arr` ($a$) can be made non-decreasing using only rotations.

**HumanEval+ canonical solution:** Enumerate the rotations of $a$ and check if it is sorted. The time complexity is $O(|a|^2)$.

```
1  sorted_arr = sorted(arr)
2  if arr == sorted_arr: return True
3  for i in range(1, len(arr)):
4    if arr[i:] + arr[:i] == sorted_arr:
5      return True
6  return False
```

**Our expert-written solution:** Note that the desired condition is equivalent to the condition that there is at most $0 \le i < |a|$ with $a_i > a_{(i+1) \bmod n}$, so we can enumerate $i$ and check this equivalent condition. The time complexity is $O(|a|)$.

```
1  if len(arr) <= 2:
2      return True
3  cnt = int(arr[-1] > arr[0])
4  for a, b in zip(arr[: -1], arr[1 :]):
5      if a > b:
6          cnt += 1
7          if cnt > 1:
8              return False
9  return True
```

### E.5   HUMANEVAL PROBLEM #154

**Problem description:** Check if any rotation of a string $b$ is a substring of a string $a$.

**HumanEval+ canonical solution:** Enumerate rotations and run brute-force string matching. The time complexity is $O(|b|^2|a|)$.

```
1  if a == b:
2      return True
3  if b == "":
4      return True
5  for i in range(0, len(b)):
6      if b[i:] + b[:i] in a:
7          return True
8  return False
```

**Our expert-written solution:** Note that the desired condition is equivalent to the condition that the longest common substring of $a$ and $b + b$ is at least $|b|$. Thus, we can run the suffix automaton of $a$ w.r.t. $b + b$ to compute their longest common substring. Since the suffix automaton of $a$ can be built within $\Theta(|a|)$ time, the overall time complexity is $O(|a| + |b|)$.

```
1  from copy import deepcopy
2  class State:
3      def __init__(self, len = 0, link = 0, next = None):
4          self.len = len
5          self.link = link
6          self.next = dict() if next is None else deepcopy(next)
7  st = [State(len = 0, link = -1)]
8  last = 0
9  def sam_extend(c, last): # to build the suffix automaton
10     cur = len(st)
11     st.append(State(len = st[last].len + 1))
12     p = last
13     while p != -1 and c not in st[p].next:
14         st[p].next[c] = cur
15         p = st[p].link
16     if p == -1:
17         st[cur].link = 0
18     else:
19         q = st[p].next[c]
20         if st[p].len + 1 == st[q].len:
21             st[cur].link = q
22         else:
23             clone = len(st)
24             st.append(State(len = st[p].len + 1, link = st[q].link, next = st[q].next))
25             while p != -1 and st[p].next[c] == q:
26                 st[p].next[c] = clone
27                 p = st[p].link
28             st[q].link = st[cur].link = clone
```

```
29    last = cur
30    return last
31  for c in a:
32    last = sam_extend(c, last)
33  v = 0
34  l = 0
35  for c in b + b:
36    while v and c not in st[v].next:
37      v = st[v].link
38      l = st[v].len
39    if c in st[v].next:
40      v = st[v].next[c]
41      l += 1
42    if l >= len(b):
43      return True
44  return False
```

