# OpenReview forum: "How efficient is LLM-generated code? A rigorous & high-standard benchmark"
_ICLR.cc/2025/Conference — ICLR 2025 Poster_

### Official Review · Reviewer_1vcL · 2024-10-26

**Soundness:** 2
**Presentation:** 3
**Contribution:** 3
**Rating:** 6
**Confidence:** 4

**Summary:**

This paper proposes a benchmark suite, named `ENAMEL`, to evaluate the efficiency of code generated by LLMs. The authors propose a novel metric eff@k that measures the efficiency of the generated code accounting for different input sizes. A human expert provides the reference implementation for each task in the benchmark suite. The evaluation with regard to the eff@k metric demonstrates the large gap between the LLM-generated code and the reference implementation.

**Strengths:**

1. Human expert-provided reference implementations for the benchmark are of high quality.
2. Evaluation provides insights into the efficiency of LLM-generated code.
3. Clear presentation that is easy to follow.

**Weaknesses:**

1. The novelty of the proposed eff@k metric is not convincing.

The authors mention the insufficiencies of existing efficiency benchmarks in C1 and C2.

a) For C1, the authors mention that if the code execution exceeds the time limit, the actual efficiency is unknown. However, the authors do not demonstrate how frequently this happens in practice. If this is a rare case, then existing benchmarks are still useful. The authors should provide experimental evidence to demonstrate that existing works do give wrong efficiency measurements due to the presence of such cases.

b) For C2, the authors argue that existing works fail to capture the relationship between code efficiency and the sample size k. By looking at Eq. (5), the proposed solution of ENAMEL is to take the maximum of the efficiency value, while the existing works take the average. The authors do not justify why taking the maximum is a better choice than taking the average. Such a difference in the metric definition seems to be a minor difference.

2. Lack of evaluation on different test case generation strategies.

The code efficiency can be affected by the testing inputs. Ideally, the efficiency should be measured against the worst-case inputs, which can be hard to define in practice. A practical approximation is to take the maximum execution time over a set of testing inputs. As such, the testing input generation is crucial for the efficiency evaluation. It is not clear how strong the test case generation strategy adopted in this paper is. The authors should evaluate the robustness of the proposed eff@k metric against different test case generation strategies.

3. Lack of guidance on setting the hyperparameters for the eff@k metric.

The eff@k metric has a set of hyperparameters, such as $\alpha$ and $h$. The value of these hyperparameters directly affects the efficiency measurement. Without guidance on how to set these hyperparameters, the practical usage of the eff@k metric is limited. It would be better if automatic hyperparameter selection methods were provided; or at least, empirical studies on how the eff@k metric is affected by different hyperparameters.

4. The benchmark suite is limited only to the HumanEval dataset.

This concerns about the generalizability of ENAMEL. The HumanEval dataset contains many simple tasks that require $\mathcal{O}(1)$ time complexity. A potential concern is that the dataset after filtering out those simple tasks may not be large enough to provide a comprehensive evaluation of the efficiency of LLM-generated code.

a) The authors should provide the size and proportion of the remaining tasks after filtering out the simple tasks.

b) The authors can consider extending the benchmark suite to more competitive datasets, such as the CodeContests dataset.

**Questions:**

1. Why is the maximum efficiency value chosen in the eff@k metric, rather than the average efficiency value?
2. Do the eff@k metric and the existing efficiency metrics give different efficiency rankings for the same LLM-generated code?
3. How sensitive is the eff@k metric to the hyperparameters?
4. What is the size of the ENAMEL benchmark suite?
5. How does the ENAMEL benchmark perform on more competitive datasets, such as the CodeContests dataset?
6. How does the eff@k metric perform on different test case generation strategies?

---

> ### Author Response · Authors · 2024-11-23
> **Response to Reviewer 1vcL (Part 1/4)**
>
> Thank you so much for your thorough review and interesting questions. We believe that your questions will greatly help to improve the quality and clarity of this paper. We answer your questions as follows.
>
> > *W1 Part a: The novelty of the proposed eff@k metric is not convincing. The authors mention the insufficiencies of existing efficiency benchmarks in C1 and C2. (a) For C1, the authors mention that if the code execution exceeds the time limit, the actual efficiency is unknown. However, the authors do not demonstrate how frequently this happens in practice. If this is a rare case, then existing benchmarks are still useful. The authors should provide experimental evidence to demonstrate that existing works do give wrong efficiency measurements due to the presence of such cases.*
> >
> > *Q2: Do the eff@k metric and the existing efficiency metrics give different efficiency rankings for the same LLM-generated code?*
>
> Thank you for the great suggestion. We have added an experiment to compare our eff metric and the most popular metric *speedup* [1] (i.e., execution time of reference solution / execution time of generated code). The table below shows rankings of LLMs under our eff metric and the speedup metric, respectively. We can see that eff and speedup give very different rankings. In particular, **under the speedup metric, Mixtral and Llama 3 even seems to outperform GPT-4**, which is inconsistent with commonsense knowledge.
>
> The unreasonable ranking by the speedup metric is because the speedup metric **overestimates efficiency** in the presence of **right-censored execution time** (i.e., when the program exceeds the time limit), as we discussed in Challenge C1.
>
> In fact, the time limits are exceeded very frequently in our evaluation, thanks to our high-quality reference solutions and challenging test cases. For example, **even for GPT-4, 16\% of its generated code exceeded the time limit** in our evaluation; for Mixtral 8x22B Instruct, 23\% of its generated code exceeded the time limit. Therefore, it is necessary to propose our eff metric to more rigorously handle right-censored execution time. We have added this experiment to Appendix B.
>
> |Rank|eff (ours)|speedup|
> |:-:|:--|:--|
> |1|GPT-4 Turbo|GPT-4 Turbo|
> |2|**GPT-4**|**Mixtral 8x22B Instruct**|
> |3|Llama 3 70B Instruct|Llama 3 70B Instruct|
> |4|**Mixtral 8x22B Instruct**|**GPT-4**|
> |5|Claude 3 Opus|Claude 3 Opus|
> |6|Phind Code Llama V2|Phind Code Llama V2|
> |7|**Claude 3 Haiku**|**ChatGPT**|
> |8|**ChatGPT**|**Claude 3 Haiku**|
> |9|Claude 3 Sonnet|Claude 3 Sonnet|
> |10|Llama 3 8B Instruct|Llama 3 8B Instruct|
> |11|**Code Llama 34B Python**|**Mixtral 8x7B Instruct**|
> |12|**Mixtral 8x7B Instruct**|**Code Llama 34B Python**|
>
> > *W1 Part b: (b) For C2, the authors argue that existing works fail to capture the relationship between code efficiency and the sample size k. By looking at Eq. (5), the proposed solution of ENAMEL is to take the maximum of the efficiency value, while the existing works take the average. The authors do not justify why taking the maximum is a better choice than taking the average. Such a difference in the metric definition seems to be a minor difference.*
> >
> > *Q1: Why is the maximum efficiency value chosen in the eff@k metric, rather than the average efficiency value?*
>
> We are afraid this is a misunderstanding. In fact, the average efficiency value over n samples is exactly **our proposed n-sample estimator for eff@1** (a special case of our Rao--Blackwellized eff@k). If one wants to find the average efficiency over n samples, then they can choose the n-sample estimator for eff@1; if one wants to find the expected best efficiency over k samples using n samples, then they can choose the n-sample estimator for eff@k.

---

> ### Author Response · Authors · 2024-11-23
> **Response to Reviewer 1vcL (Part 2/4)**
>
> > *W2: Lack of evaluation on different test case generation strategies. The code efficiency can be affected by the testing inputs. Ideally, the efficiency should be measured against the worst-case inputs, which can be hard to define in practice. A practical approximation is to take the maximum execution time over a set of testing inputs. As such, the testing input generation is crucial for the efficiency evaluation. It is not clear how strong the test case generation strategy adopted in this paper is. The authors should evaluate the robustness of the proposed eff@k metric against different test case generation strategies.*
> >
> > *Q6: How does the eff@k metric perform on different test case generation strategies?*
>
> Thank you for the great question. In fact, the strength of our test case generators have been ensured by our human expert. Even though worst cases might be hard to identify for real-world problems, it is **not hard for our human expert to identify worst cases for HumanEval problems** because these problems are well defined. As discussed in Section 3.3, our human expert will add worst cases for any possible algorithm, **including non-trivial worst cases**. For example, for problem \#31 (deciding if a number is prime), the Fermat primality test (generated by Code Llama 13B Python) is an **efficient yet wrong** algorithm with only a few non-trivial counterexamples called *Carmichael numbers* (Carmichael, 1912).
>
> To further demonstrate the strength of our expert-written test case generators, we provide a case study comparing our strong generator and the random generator for the problem \#31 mentioned above. We compare the eff@1 metrics under our test case generator and the random generator for two solutions ((i) Factor: the naive factorization algorithm, which is correct but inefficient; (ii) Fermat: the Fermat primality test, which is efficient but wrong), respectively. The results are shown in the table below. We can see that **random test cases cannot assess true correctness or true efficiency** while our test case generator can. This demonstrates the strength of our expert-written test case generators. We have added this experiment to Appendix B.
>
> |Generator|Factor|Fermat|
> |:--|:-:|:-:|
> |Random|0.91|1.25|
> |**Ours**|**0.17**|**0.00**|
>
> > *W3: Lack of guidance on setting the hyperparameters for the eff@k metric. The eff@k metric has a set of hyperparameters, such as $\alpha$ and $h$. The value of these hyperparameters directly affects the efficiency measurement. Without guidance on how to set these hyperparameters, the practical usage of the eff@k metric is limited. It would be better if automatic hyperparameter selection methods were provided; or at least, empirical studies on how the eff@k metric is affected by different hyperparameters.*
> >
> > *Q3: How sensitive is the eff@k metric to the hyperparameters?*
>
> Thank you for the great question. Regarding alpha, alpha represents the tolerence to execution time (i.e., the execution time limit is proportional to alpha). Thus, if one wants to tolerate less efficient code, then they can use a larger alpha. Regarding h, it represents how we weigh each level. Thus, if one wants to focus more on easier levels, they should use a larger h1; if one wants to focus more on harder levels, they should use a larger h3. **We encourage users to stick to our default hyperparameters** alpha=2, h1=3, h2=3, h3=4 to ensure consistency across different test cases and different LLMs. We used these default hyperparameters throughout this work.
>
> To further illustrate how eff@k is affected by alpha and h, we report the eff@1 of GPT-4 Turbo under different alpha, h1, h2, and h3. Results are shown in the tables below. We can see that eff@1 increases as alpha increases (because alpha represents the tolerence to less efficient code), that eff@1 increases as h1 increases (because we weigh more on an easier level), and that eff@1 decreases as h2 or h3 increases (because we weigh more on a harder levels). These empirical results are consistent with the aforementioned analysis. We hope these empirical results will help users decide hyperparameters based on their preferences in terms of the tolerance to execution time vs. weights across different levels. We have added this experiment to Appendix B.
>
> |alpha|1.5|2.0|2.5|3.0|3.5|
> |:-:|:-:|:-:|:-:|:-:|:-:|
> |eff@1|0.421|0.470|0.502|0.525|0.541|
>
> |h1|1|2|3|4|5|
> |:-:|:-:|:-:|:-:|:-:|:-:|
> |eff@1|0.428|0.451|0.470|0.486|0.498|
>
> |h2|1|2|3|4|5|
> |:-:|:-:|:-:|:-:|:-:|:-:|
> |eff@1|0.474|0.472|0.470|0.469|0.467|
>
> |h3|1|2|3|4|5|
> |:-:|:-:|:-:|:-:|:-:|:-:|
> |eff@1|0.520|0.499|0.483|0.470|0.460|

---

> ### Author Response · Authors · 2024-11-23
> **Response to Reviewer 1vcL (Part 3/4)**
>
> > *W4 Part a: The benchmark suite is limited only to the HumanEval dataset. This concerns about the generalizability of ENAMEL. The HumanEval dataset contains many simple tasks that require time complexity. A potential concern is that the dataset after filtering out those simple tasks may not be large enough to provide a comprehensive evaluation of the efficiency of LLM-generated code. (a) The authors should provide the size and proportion of the remaining tasks after filtering out the simple tasks.*
> >
> > *Q4: What is the size of the ENAMEL benchmark suite?*
>
> We would like to provide the following clarificaitons. As mentioned in Section 1 and Section 3.1, our benchmark has **142** problems, which is **87\%** size of HumanEval (164 problems). In fact, only 22 problems in HumanEval have O(1) time complexity; all other 142 problems have higher time complexities. In other words, our benchmark has a **comparable size to HumanEval.** Our human expert has manually inspected all these 142 problems. % which makes our benchmark a valuable resource for efficiency evaluation due to the following reasons.
>
> Despite the size of the benchmark, **our 142 problems has already revealed the limited capability** of all the 30 LLMs in generating efficient code. In particular, our benchmark shows that even the strongest LLM (GPT-4 Turbo) is still far from generating expert-level efficient code (eff@1 below 0.5). We hope our findings and benchmark will help LLM developers to realize this critical issue and further inspire them to develop stronger LLM code generators. The effectiveness of our benchmark is because **our human expert has carefully verified the comprehensiveness** of the problemset:
>
>   - (i) As shown in Figure 2, our benchmark problems **have diverse difficulties**. For example, 75 seemingly easy problems require non-trivial implementation optimization, and 20 hard problems require advanced algorithms.
>   - (ii) As discussed in Section 3.2, our problemset **covers a wide range of algorithmic knowledge** (including data structures, dynamic programming, and automata) and **a wide range of mathematical knowledge** (including linear algebra, combinatorics, and number theory).
>
> Therefore, we believe that our 142 problems offer a comprehensive evaluation of LLMs' capability in generating efficient code.
>
> We believe that crafting the benchmark by **human expert is necessary** to develop a **high-standard and rigorous** benchmark. For example, as shown in Table 2, compared with our expert solutions, HumanEval canonical solutions achieve only eff@1=0.455, and HumanEvalPlus canonical solutions achieve only eff@1=0.513. This shows that their canonical solutions are **far less efficient than our expert-written reference solutions**.
>
>   - (i) We did not use problems/solutions from **online judges** (like LeetCode or Codeforces) because their **public solutions are already in LLMs' pretraining corpuses**. For example, DeepMind's AlphaCode (Li et al., 2022) has been trained on many online judges including Codeforces, CodeChef, HackerEarth, AtCoder, and Aizu. If we evaluate LLMs on these public online judges, then the evaluation results may fail to reflect the LLMs’ true capabilities due to test set leakage / memorization.
>   - (ii) We did not **crowd-source** the benchmark because otherwise it would be **hard to guarantee the quality** of the benchmark. For example, MBPP (Austin et al., 2021) is a popular crowd-sourced benchmark, but it is known to be easier than HumanEval (Rozière et al., 2023).
>   - (iii) We did not use **LLM-generate reference solutions** because LLM-generated code are still far from expert-level efficiency, as demonstrated in our evaluation.
>
> That said, we agree with the reviewers that addressing scalablity of benchmark development is an important future direction. A possible solution is to **collaborate with private programming competitions** whose solutions are not publicly available. Nevertheless, we believe that this will be an issue *only after* LLMs can generate efficient code on our benchmark but might not for now. We have added a discussion to Appendix D.

---

> ### Author Response · Authors · 2024-11-23
> **Response to Reviewer 1vcL (Part 4/4)**
>
> > *W4 Part b: (b) The authors can consider extending the benchmark suite to more competitive datasets, such as the CodeContests dataset.*
> >
> > *Q5: How does the ENAMEL benchmark perform on more competitive datasets, such as the CodeContests dataset?*
>
> Thank you for the great question. As mentioned above, our main concern about using online judge datasets like CodeContests is that **their public solutions are already in LLMs' pretraining corpuses**. For example, DeepMind's AlphaCode (Li et al., 2022) **has been trained on the CodeContests dataset**. If we evaluate LLMs on these public online judges, then the evaluation results may fail to reflect the LLMs' true capabilities due to potential memorization.
>
> To evaluate LLMs' true capabilities, we decided to employ human expert to manually write our reference solutions to ensure a rigorous evaluation. As a result, we have successfully revealed that even the strongest LLMs (e.g., GPT-4) still fall short of generating expert-level efficient solutions.
>
> Nevertheless, we with the reviewer in that extending the benchmark suite to more competitive datasets will be a very interesting future direction. A possible workaround is to collaborate with competitions whose solutions are not publicly available. We have added a discussion to Appendix D.
>
> ---------
>
> References:
>
> - Carmichael. On composite numbers p which satisfy the Fermat congruence. The American Mathematical Monthly, 19(2):22-27, 1912.
> - Li et al. Competition-level code generation with AlphaCode. Science, 378(6624):1092-1097, 2022.
> - Austin et al. Program synthesis with large language models. Google Research, 2021.
> - Rozière et al. Code Llama: Open foundation models for code. Meta AI, 2023.

---

> > ### Comment · Reviewer_1vcL · 2024-11-26
> >
> > Hello, authors, thanks for your clarifications. The contributions of this paper still look vague to me.
> >
> > ## Questions about the contribution of proposing a new benchmark on HumanEval
> >
> > One of the contributions is to propose a new efficiency benchmark on HumanEval. However, there are already some efficiency benchmarks on such datasets, like the following:
> >
> > 1. [Effibench: Benchmarking the efficiency of automatically generated code](https://arxiv.org/abs/2402.02037)
> > 2. [Mercury: A code efficiency benchmark for code large language models](https://openreview.net/forum?id=vyraA7xt4c)
> >
> > The Effibench focuses on competitive programming contests like LeetCode and contains 1,000 cases. The Mercury benchmark, and its improved version, Venus, contain over 1,800 tasks with diverse difficulty levels and support multiple languages.
> >
> > Both benchmarks contain significantly more tasks than ENAMEL, which only contains 142 cases. Also, ENAMEL only supports Python. HumanEval is also much simpler than LeetCode, which makes it less challenging for LLMs to solve.
> >
> > The authors then argue that competitive programming contests can have data leakage issues. In spite of that, it is possible that LeetCode may still be a reasonable benchmark for efficiency evaluation purposes. LLMs may not have seen the optimal solutions to LeetCode. According to the results of Effibench, LLMs have difficulties producing optimal solutions for LeetCode, e.g., GPT-4 generated code is 3.12 times slower than the canonical solutions.
> >
> > The authors also suggest collaborating with private programming contests. I am not convinced that such a suggestion is easy to scale up. A single private programming contest contains merely a dozen programming problems. Due to the scarcity of such contests, it can be hard to obtain over a thousand programming problems.
> >
> > ## Questions about the contribution of the proposal of the eff@k metric
> >
> > Another contribution is the proposal of the eff@k metric. In the follow-up clarifications, the authors demonstrate that traditional efficiency metrics like speedup suffer from the issue of right-censored execution time and, thus are less suitable for efficiency evaluation. The authors also show that the issue of right-censored execution time is rather frequent, with 16% of GPT-4 generated code exceeding the time limit.
> >
> > However, there are three issues with the eff@k metric:
> >
> > 1. It is less intuitive to interpret the results. For the traditional speedup metric, it is easy to say that program A is xxx times faster than program B. By looking at the values of the eff@k metric, can we draw other conclusions besides the fact that program A is faster than program B?
> > 2. The issue of right-censored execution time depends on the time limit. In the case of severe right-censored time issues, it is possible that the time limit is not set correctly. Why not increase the time limit, or skip test cases that take extremely long to finish?
> > 3. The evaluation of the eff@k metric is only performed on the HumanEval benchmark, thus challenging its generalizability. It would be better if the authors could evaluate the utility of the eff@k metric on other benchmarks, such as the Effibench and Mercury.

---

> > > ### Author Response · Authors · 2024-11-29
> > > **Follow-up response (Part 1/2)**
> > >
> > > Thank you for your detailed follow-up questions. We provide our response below.
> > >
> > > > *The Effibench focuses on competitive programming contests like LeetCode and contains 1,000 cases. The Mercury benchmark, and its improved version, Venus, contain over 1,800 tasks with diverse difficulty levels and support multiple languages. Both benchmarks contain significantly more tasks than ENAMEL, which only contains 142 cases.*
> > >
> > > Despite the size of the benchmark, our 142 problems **has already revealed the limited capability** of all the 30 LLMs in generating efficient code. In particular, our benchmark shows that even the strongest LLM (GPT-4 Turbo) is still far from generating expert-level efficient code (eff@1 below 0.5). We agree that developing a larger dataset is an important future direction, but that will
> > > be an issue **only after LLMs can generate efficient code on our benchmark** but might not be an issue for now.
> > >
> > > > *HumanEval is also much simpler than LeetCode, which makes it less challenging for LLMs to solve. The authors then argue that competitive programming contests can have data leakage issues. In spite of that, it is possible that LeetCode may still be a reasonable benchmark for efficiency evaluation purposes. LLMs may not have seen the optimal solutions to LeetCode.*
> > >
> > > To further demonstrate the difficulty of our problems, we provide a comparison of evaluation results between EffiBench, Mercury, and our benchamrk ENAMEL. Since Mercury did not evaluate GPT-4 or CodeLlama 70B, we compare the reported evaluation results of CodeLlama 34B. The results are shown below. From the table, we can see that EffiBench's and Mercury's **LeetCode problems are less challenging to LLMs than our benchmark**. This is because LLMs (i) **have seen the public solutions on LeetCode** (editorials and community solutions) but (ii) **have never seen our expert-written efficient solutions**.
> > >
> > > |Benchmark|Metric|Result|
> > > |:--|:-:|:--|
> > > |EffiBench|1/NET|0.336|
> > > |Mercury|Beyond|0.424|
> > > |**ENAMEL (ours)**|**eff@1**|**0.268**|
> > >
> > > (Unfortunately, we were unable to find a description of the Venus dataset. Their GitHub repo is **still work in progress**. Could you provide a reference for it or provide their results for CodeLlama 34B?)
> > >
> > > > *According to the results of Effibench, LLMs have difficulties producing optimal solutions for LeetCode, e.g., GPT-4 generated code is 3.12 times slower than the canonical solutions.*
> > >
> > > We would like to point out that **the 3.12x slowdown by GPT-4 is in fact small**, which demonstrates that **data leakage** has indeed contributed to GPT-4's performance on LeetCode. In contrast, on our benchmark, even the strongest LLM GPT-4 Turbo generates drastically slower code than our expert solutions. For example,
> > >
> > > - For HumanEval \#36, GPT-4 Turbo's code is **3497** times slower than our expert solution (ours: 0.03 s; GPT-4 Turbo: 95.09 s);
> > > - For HumanEval \#55, GPT-4 Turbo's code is **33** times slower than our expert solution (ours: 0.06 ms; GPT-4 Turbo: 2.03 ms);
> > > - For HumanEval \#89, GPT-4 Turbo's code is **179** times slower than our expert solution (ours: 0.05 ms; GPT-4 Turbo: 8.66 ms);
> > > - For HumanEval \#103, GPT-4 Turbo's code is **6878896** times slower than our expert solution (ours: 0.0000009 s; GPT-4 Turbo: 6.23 s);
> > > - For HumanEval \#107, GPT-4 Turbo's code is **18426582** times slower than our expert solution (ours: 0.00002 s; GPT-4 Turbo: 301.38 s).

---

> > > ### Author Response · Authors · 2024-11-29
> > > **Follow-up response (Part 2/2)**
> > >
> > > > *The authors also suggest collaborating with private programming contests. I am not convinced that such a suggestion is easy to scale up. A single private programming contest contains merely a dozen programming problems. Due to the scarcity of such contests, it can be hard to obtain over a thousand programming problems.*
> > >
> > > In fact, there are abundant private contests. One example is the [ACM International Collegiate Programming Contest (ICPC)](https://icpc.global/regionals/upcoming). ACM has been holding a lot of ICPC regional contests all over the world (including Asia--Pacific, Africa--Arabia, Asia West, Northern Eurasia, North America, Latin America, Europe, and Asia East) **every year since 1998**.
> > >
> > > For instance, in 2023, ACM has held **329** ICPC regional contests, and each contest has 8-12 problems on average. Thus, even if we only use the ICPC regional contests in 2020-2023, we will already be able to create a dataset of **over 10000 problems** with expert-written reference solutions and strong test cases.
> > >
> > > > *Q7: It is less intuitive to interpret the results. For the traditional speedup metric, it is easy to say that program A is xxx times faster than program B. By looking at the values of the eff@k metric, can we draw other conclusions besides the fact that program A is faster than program B?*
> > >
> > > Sorry for the confusion. Our eff metric is just **a linear transformation of the speedup metric** when the execution does not exceed the time limit. Hence, it can be interpreted as **a generalization of the speedup metric** for right-cencored execution time.
> > >
> > > Let us elaborate its connection with speedup. Let $\alpha>1$ be the timeout factor, $t^\*>0$ be the execution time of the reference solution, and $0<t<\alpha t^\*$ be the true execution time of the generated code. Then, $speedup=\frac{t}{t^\*}$, and our eff metric is $\frac{\alpha t^\*-t}{\alpha t^\*-t^\*}=\frac{\alpha}{\alpha-1}-\frac{1}{\alpha-1}\cdot\frac{t}{t^\*}=\frac{\alpha}{\alpha-1}-\frac{speedup}{\alpha-1}$. We can see that our eff metric is a linear transformation of speedup.
> > >
> > > For example, if $\alpha=3$ and eff=0.8, then we can interpret that the generated code is 1.4 times slower than the reference solution.
> > >
> > > > *Q8: The issue of right-censored execution time depends on the time limit. In the case of severe right-censored time issues, it is possible that the time limit is not set correctly. Why not increase the time limit, or skip test cases that take extremely long to finish?*
> > >
> > > Sorry for the confusion. We would like to clarify that the time limit in our benchmark represents the user's tolerance to inefficient solutions, and **our eff metric represents the LLM's capability to satisfy the user's tolerance**. Therefore, **the time limit is mainly up to user's preference**. If the user wants to tolerate less efficient code, then they can use a larger timeout factor $\alpha$, and vice versa.
> > >
> > > > *Q9: The evaluation of the eff@k metric is only performed on the HumanEval benchmark, thus challenging its generalizability. It would be better if the authors could evaluate the utility of the eff@k metric on other benchmarks, such as the Effibench and Mercury.*
> > >
> > > Sorry for the confusion. As described in Section 2, our eff@k metric is applicable as long as the benchmark provides **high-quality** reference solutions and test case generators. As shown by our results above, EffiBench and Mercury are less challenging to LLMs than our benchmark, so they might not have expert-level efficient solutions for many problems. Hence, we believe that our benchmark is more suitable to use the eff@k metric than EffiBench and Mercury.

---

> > > > ### Comment · Reviewer_1vcL · 2024-12-01
> > > >
> > > > Thanks for your detailed response. The answers look reasonable to me. I would be happy to increase the ratings.
> > > >
> > > > Regarding your question:
> > > > > Unfortunately, we were unable to find a description of the Venus dataset
> > > >
> > > > They have released their dataset on GitHub and HuggingFace. You can find the Venus dataset in the following links:
> > > > 1. https://github.com/Elfsong/venus
> > > > 2. https://huggingface.co/datasets/Elfsong/Venus

---

> > > > > ### Author Response · Authors · 2024-12-01
> > > > >
> > > > > Thank you so much for your recognition and for the reference to the Venus dataset!

---

### Official Review · Reviewer_2JoC · 2024-11-03

**Soundness:** 3
**Presentation:** 4
**Contribution:** 2
**Rating:** 6
**Confidence:** 4

**Summary:**

The paper presents ENAMEL, a benchmark for evaluating the efficiency of LLM-generated code, a quality largely overlooked in current assessments. ENAMEL introduces a new efficiency metric, eff@k, and includes expert-designed reference solutions and rigorous test cases to set high standards. Evaluating 30 popular LLMs, the authors find that while many models generate correct code, they fall short in efficiency, particularly with advanced algorithms and optimizations, underscoring the need for further improvements in LLM code synthesis.

**Strengths:**

1. Clear and fluent presentation. The metric is carefully designed.
2. The evaluation is convincing and indicative.

**Weaknesses:**

1. The method is very manual. It is hard to scale.
2. The dataset is small. Humaneval was a good metric several years ago at the early stage of code generation. But now, serious evaluation needs a larger scale dataset.

Despite the above, the dataset is still reasonably valuable due to its high quality.

**Questions:**

1. Why didn't you consider problems from online judges like codeforces? There are plenty of problems with expert-level efficient solutions as a competition-oriented platform.
2. Did you consider a more automatic way of measuring the complexity? It might not be hard to find out that multiple layers of input scale, simply considering log scales, polynomial scales, and exponential scales.

---

> ### Author Response · Authors · 2024-11-23
> **Response to Reviewer 2JoC (Part 1/2)**
>
> Thank you so much for taking the time to provide such an insightful and inspiring review. We highly appreciate that you recognize the clear presentation of this paper, our rigorously design efficiency metric, and our convincing evaluation results. We believe that your comments will greatly improve the quality of this paper. We provide our response below.
>
> > *W1: The method is very manual. It is hard to scale.*
>
> As discussed in Appendix D, we agree that our current expert-based approach has a scalability limitation because we primarily focus on **ensuring a high-quality dataset**. We believe that crafting the benchmark by **human expert is necessary** to develop a **high-standard and rigorous** benchmark. For example, as shown in Table 2, compared with our expert solutions, HumanEval canonical solutions achieve only eff@1=0.455, and HumanEvalPlus canonical solutions achieve only eff@1=0.513. This shows that their canonical solutions are **far less efficient than our expert-written reference solutions**.
>
>   - (i) We did not use problems/solutions from **online judges** (like LeetCode or Codeforces) because their **public solutions are already in LLMs' pretraining corpuses**. For example, DeepMind's AlphaCode [1] has been trained on many online judges including Codeforces, CodeChef, HackerEarth, AtCoder, and Aizu. If we evaluate LLMs on these public online judges, then the evaluation results may fail to reflect the LLMs’ true capabilities due to test set leakage / memorization.
>   - (ii) We did not **crowd-source** the benchmark because otherwise it would be **hard to guarantee the quality** of the benchmark. For example, MBPP [2] is a popular crowd-sourced benchmark, but it is known to be easier than HumanEval [3].
>   - (iii) We did not use **LLM-generate reference solutions** because LLM-generated code are still far from expert-level efficiency, as demonstrated in our evaluation. For example, as shown in Table 2, compared with our expert solutions, HumanEval canonical solutions achieve only eff@1=0.455, and HumanEvalPlus canonical solutions achieve only eff@1=0.513. This shows that their canonical solutions are **far less efficient than our expert-written reference solutions**.
>
> That said, we still agree with the reviewer that addressing scalablity of benchmark development is an important future direction. A possible solution is to **collaborate with private programming competitions** whose solutions are not publicly available. %Nevertheless, this will be an issue *only after* LLMs can generate efficient code on our benchmark.
> We have added a discussion to Appendix D.
>
> > *W2: The dataset is small. Humaneval was a good metric several years ago at the early stage of code generation. But now, serious evaluation needs a larger scale dataset.*
>
> Thank you for the great suggestion. We agree that the dataset size is a limitation of this work that we hope to address in future work. Meanwhile, despite the size of the benchmark, **our 142 problems has already revealed the limited capability** of all the 30 LLMs in generating efficient code. In particular, our benchmark shows that even the strongest LLM (GPT-4 Turbo) is still far from generating expert-level efficient code (eff@1 below 0.5). We hope our findings and benchmark will help LLM developers to realize this critical issue and further inspire them to develop stronger LLM code generators. The effectiveness of our benchmark is because **our human expert has carefully verified the comprehensiveness** of the problemset:
>
>   - (i) As shown in Figure 2, our benchmark problems **have diverse difficulties**. For example, 75 seemingly easy problems require non-trivial implementation optimization, and 20 hard problems require advanced algorithms.
>   - (ii) As discussed in Section 3.2, our problemset **covers a wide range of algorithmic knowledge** (including data structures, dynamic programming, and automata) and **a wide range of mathematical knowledge** (including linear algebra, combinatorics, and number theory).
>
> That said, we still agree with the reviewers that developing a larger benchmark is an important future direction. As we mentioned in W1, a possible solution is to **collaborate with private programming competitions** whose solutions are not publicly available. We have added a discussion to Appendix D. Thank you again for the great suggstion.

---

> ### Author Response · Authors · 2024-11-23
> **Response to Reviewer 2JoC (Part 2/2)**
>
> > *W3: Why didn't you consider problems from online judges like codeforces? There are plenty of problems with expert-level efficient solutions as a competition-oriented platform.*
>
> Thank you for the great question. As mentioned in W1, our main concern about using online judges is that **their public solutions are already in LLMs' pretraining corpuses**. For example, DeepMind's AlphaCode [1] has been trained on many online judges **including Codeforces**, CodeChef, HackerEarch, Aizu, and AtCoder. If we evaluate LLMs on these public online judges, then the evaluation results may fail to reflect the LLMs' true capabilities due to potential memorization.
>
> To evaluate LLMs' true capabilities, we decided to employ human expert to manually write our reference solutions. As a result, we have successfully revealed that even the strongest LLMs (e.g., GPT-4) still fall short of generating expert-level efficient solutions.
>
> Nevertheless, we still agree with the reviewer that extending the benchmark suite to more competitive datasets will be a very interesting future direction. As we mentioned in W1, a possible workaround is to collaborate with competitions whose solutions are not publicly available. We have added a discussion to Appendix D.
>
> > *W4: Did you consider a more automatic way of measuring the complexity? It might not be hard to find out that multiple layers of input scale, simply considering log scales, polynomial scales, and exponential scales.*
>
> Thank you for the interesting question. In this work, we focus not only on time complexity **but also on implementation optimization**. Even though two solutions have the same complexity, the optimized implementation can sometimes significantly outperform the non-optimized one. For example, for problem \#98 (counting uppercase vowels at even indices), an efficient Python implementation needs a clever use of the builtin function `str.translate` rather than straightforward counting. As shown in Table 4, LLMs still fall short of generating efficient code for seemingly easy problems (the Implementation Optimization Subset). Therefore, we did not use any approach to measure the time complexity only.
>
> That said, we agree with the reviewer in that developing an automatic method to measure the time complexity will be a very intersting future work. Although this might require an indepdent new study, there are two possible approaches (although both of them have limitations):
>
> - **Time complexity prediction:** A possible approach is to train an LLM to predict the time complexity of a given code sample. However, existing time complexity analyzers (such as LeetCode's analyzer) are known to be inaccurate. We believe that time complexity prediction is in general difficult for LLMs (and even diffcult for non-expert humans). For example, the Splay tree [4] **seems to have O(n) time complexity** per operation, but a sophisticated analysis by the authors shows that it actually has **O(log n) time complexity** per operation.
> - **Fitting a time curve:** Another possible approach is to fit a curve of the running time vs $n$ to help decide the time complexity. However, we believe that this is in general difficult because it is practically infeasible to distinguish a high-degree polynomial from an exponential function. For example, the AKS primality test [5] runs in $\tilde O((\log n)^{12})$ time, so the curve of its running time vs $n$ **looks extremely like an exponential function** for any practical $n$ (it cannot even decide whether $n=50$ is a prime number within reasonable time).
>
> We have added a discussion on automatic measurement of time complexity into Appendix D.
>
> ----------------------------
>
> References:
>
> - [1] Li et al. Competition-level code generation with AlphaCode. Science. 378(6624):1092-
> 1097, 2022.
> - [2] Austin et al. Program synthesis with large language models. Google Research, 2021.
> - [3] Rozière et al. Code Llama: Open foundation models for code. Meta AI, 2023.
> - [4] Sleator et al. Self-adjusting binary search trees. Journal of the ACM, 1985.
> - [5] Agrawal et al. PRIMES is in P. Annals of Mathematics. 160(2):781–793, 2004.

---

### Official Review · Reviewer_t6vW · 2024-11-03

**Soundness:** 3
**Presentation:** 3
**Contribution:** 2
**Rating:** 5
**Confidence:** 3

**Summary:**

The paper proposes a new benchmark for evaluating the efficiency of code generated by LLMs. The authors select 142 problems out of the 164 problems in HumanEval and HumanEval+ (they excluded trivial problems with Θ(1) time complexity). They propose a metric that they call eff@k which is the equivalent for pass@k but adapted to the problem of evaluating the performance of code. They then evaluate 30 LLMs (closed and open-source LLMs) on their benchmark using the eff@k metric. They show that state-of-the-art LLMs achieve an eff@1 score of 0.47 and an eff@100 score of 0.575. They also propose a hand-optimized version of each of the 142 problems in their benchmark to set a new standard for LLMs.

**Strengths:**

-	Evaluating whether LLMs can generate efficient code is an important problem, especially with the increasing amount of research on the topic of code synthesis using LLMs and the wide use of LLMs for coding.
-	The authors provide a comprehensive evaluation of 30 LLMs using the proposed benchmark and more importantly, provide a hand-optimized version of each one of the codes in their benchmark which opens the door for more research on the topic.

**Weaknesses:**

While the proposed work is important, and while the results are interesting, I don’t believe the contribution and novelty of the work are strong enough. There is clearly a degree of novelty in this paper though, and the proposed benchmark is very valuable from a practical point of view.

The problem of measuring the performance of code is well-studied in the compiler community. There are decades of work on the development of methods for automatic code optimization in compilers and methods to measure the success of such methods. Any paper about automatic code optimization. Here is one example of such work:

       “Towards a Statistical Methodology to Evaluate Program Speedups and their Optimisation Techniques”, Sid Touati.

It would be interesting to include examples of such work in the related work section and also include a detailed discussion in the paper about why a simple metric such as “the speedup” (ratio between the execution time of the reference code over the execution time of the synthesized code) is not enough for this problem and a new metric needs to be proposed.

**Questions:**

-	Can you please include a detailed discussion of why the the speedup metric classically used by the compiler community when evaluating automatic code optimization methods is not enough for your task?
-	For more completeness, can you also include a discussion of such work in the related work section? A starting point would be:  “Towards a Statistical Methodology to Evaluate Program Speedups and their Optimisation Techniques”, Sid Touati.

---

> ### Author Response · Authors · 2024-11-23
> **Response to Reviewer t6vW**
>
> Thank you for your great suggestions. We are grateful that you appreciate the importance of efficiency evaluation, the comprehensiveness of our evaluation, and the high quality of our expert-written reference solutions. We believe that your suggestions will considerably improve the clarity of this paper. We provide our response below.
>
> > *W1: While the proposed work is important, and while the results are interesting, I don’t believe the contribution and novelty of the work are strong enough. There is clearly a degree of novelty in this paper though, and the proposed benchmark is very valuable from a practical point of view. The problem of measuring the performance of code is well-studied in the compiler community. There are decades of work on the development of methods for automatic code optimization in compilers and methods to measure the success of such methods. Any paper about automatic code optimization. Here is one example of such work: “Towards a Statistical Methodology to Evaluate Program Speedups and their Optimisation Techniques”, Sid Touati. It would be interesting to include examples of such work in the related work section and also include a detailed discussion in the paper about why a simple metric such as “the speedup” (ratio between the execution time of the reference code over the execution time of the synthesized code) is not enough for this problem and a new metric needs to be proposed.*
> >
> > *Q1: Can you please include a detailed discussion of why the the speedup metric classically used by the compiler community when evaluating automatic code optimization methods is not enough for your task?*
>
> Thanks for the great question. As we discussed in Challenge C1, when code execution is early terminated due to time limit, its **actual execution time is unknown**. For instance, if the generated code contains an infinite loop, the observed execution time will be **clipped to the time limit**, while the actual execution time should be **infinity**. We cannot use the actual execution time in the speedup metric here because the actual execution time is unknown. Meanwhile, if we directly use the clipped execution time in the speedup metric, it will overestimate the efficiency of the code. That is why we use our eff metric instead of speedup in our evaluation.
>
> To empirically justify our eff metric, we further added an experiment to compare our eff metric and the speedup metric. The table below shows rankings of LLMs under our eff metric and the speedup metric, respectively. We can see that eff and speedup give very different rankings. In particular, **under the speedup metric, Mixtral and Llama 3 even seems to outperform GPT-4**, which is inconsistent with commonsense knowledge.
>
> The unreasonable ranking by the speedup metric is because the speedup metric **overestimates efficiency** in the presence of **right-censored execution time** (i.e., when the program exceeds the time limit), as we discussed in Challenge C1.
>
> In fact, the time limits are exceeded very frequently in our evaluation, thanks to our high-quality reference solutions and challenging test cases. For example, **even for GPT-4, 16\% of its generated code exceeded the time limit** in our evaluation; for Mixtral 8x22B Instruct, 23\% of its generated code exceeded the time limit. Therefore, it is necessary to propose our eff metric to more rigorously handle right-censored execution time. We have added this experiment to Appendix B.
>
> |Rank|eff (ours)|speedup|
> |:-:|:--|:--|
> |1|GPT-4 Turbo|GPT-4 Turbo|
> |2|**GPT-4**|**Mixtral 8x22B Instruct**|
> |3|Llama 3 70B Instruct|Llama 3 70B Instruct|
> |4|**Mixtral 8x22B Instruct**|**GPT-4**|
> |5|Claude 3 Opus|Claude 3 Opus|
> |6|Phind Code Llama V2|Phind Code Llama V2|
> |7|**Claude 3 Haiku**|**ChatGPT**|
> |8|**ChatGPT**|**Claude 3 Haiku**|
> |9|Claude 3 Sonnet|Claude 3 Sonnet|
> |10|Llama 3 8B Instruct|Llama 3 8B Instruct|
> |11|**Code Llama 34B Python**|**Mixtral 8x7B Instruct**|
> |12|**Mixtral 8x7B Instruct**|**Code Llama 34B Python**|
>
> > *Q2: For more completeness, can you also include a discussion of such work in the related work section? A starting point would be: “Towards a Statistical Methodology to Evaluate Program Speedups and their Optimisation Techniques”, Sid Touati.*
>
> Thank you for pointing out this interesting future work. We have added a discussion of this paper and related papers into Section 5.

---

> ### Author Response · Authors · 2024-12-02
> **A gentle reminder**
>
> Dear Reviewer t6vW,
>
> Apologies for the follow-up reminder.
>
> As **today (Dec 2) is the last day** for reviewers to post follow-up comments, we would like to confirm whether our responses have adequately addressed your concerns. Please let us know if you have any additional questions or require further clarifications—we would be very happy to answer them.
>
> Thank you for your time and feedback.

---

### Official Review · Reviewer_xNv8 · 2024-11-03

**Soundness:** 4
**Presentation:** 4
**Contribution:** 3
**Rating:** 6
**Confidence:** 4

**Summary:**

The paper presents a systematic benchmark for efficiency of LLM generated code. The paper presents the eff@k metric, which captures efficiency of generated code (the expected maximum efficiency score of $k$ independent code sample). The presented metric is the first that can capture the relationship between code efficiency and the sample size k. Expert solutions and expert test cases make this benchmark a compelling baseline.

**Strengths:**

- Very nice and clear presentation
- A clear and rigorous technical treatment of evaluation timeout (right censoring)
- Human expert solutions as an efficiency target
- Human expert test cases covering tricky corner cases

**Weaknesses:**

- I would appreciate a comparison with previous metrics, even the naive ones, to see whether the empirical effect of this new metric is that models/problems are given a different ranking.
- Not evaluated with a prompt that asks for an efficient solution, I suspect that a better prompt would make this entire suite too weak for being a competitive benchmark

**Questions:**

- I understand the theoretical importance of Rao-Blackwellization in this case, but what is the empirical effect?

- Effibench presented an efficiency-benchmark with 1,000 coding problems, where ach problem is paired with an executable human-written canonical solution, which obtains the SOTA efficiency on the LeetCode solution leaderboard. I would not classify it as a sporadic attempt, what makes you view it as such?

- Does it make sense to also consider space complexity in your benchmarks? For example:

Your optimized solution for #40:
```
def has_triplet_sum_zero(l):
	n = len(l)
	if n < 3:
		return False
	for i, x in enumerate(l[: n - 2]):
		buf = set()
		for y in l[i + 1 :]:
			if y in buf:
				return True
		buf.add(-x - y)
	return False
```

Suggested solution:
```
def has_triplet_sum_zero(l):
    l.sort()  # Sort the list first
    n = len(l)

    for i in range(n - 2):
        # Use two pointers to find if there's a pair with sum -l[i]
        left, right = i + 1, n - 1
        target = -l[i]

        while left < right:
            current = l[left] + l[right]
            if current == target:
                return True
            elif current < target:
                left += 1
            else:
                right -= 1

    return False
```

Uses two pointers, has the same time complexity, and reduces space complexity from $O(n)$ to $O(1)$.

---

> ### Author Response · Authors · 2024-11-23
> **Response to Reviewer xNv8 (Part 1/4)**
>
> Thank you so much for your time to provide a detailed and insightful review. We are delighted that you appreciate the clear presentation of this paper, our rigorous treatment of execution timeout, and our high-quality reference solutions and test case generators. Besides that, we believe that your questions will greatly help to improve this paper. We answer your questions as follows.
>
> > *W1: I would appreciate a comparison with previous metrics, even the naive ones, to see whether the empirical effect of this new metric is that models/problems are given a different ranking.*
>
> Thank you for the great suggestion. Per reviewer's request, we have added an additional experiment to compare our eff metric and the most popular metric *speedup* (Amdahl, 1967; i.e., execution time of reference solution / execution time of generated code). The table below shows rankings of LLMs under our eff metric and the speedup metric, respectively. We can see that eff and speedup give very different rankings. In particular, **under the speedup metric, Mixtral and Llama 3 even seems to outperform GPT-4**, which is inconsistent with commonsense knowledge.
>
> The unreasonable ranking by the speedup metric is because the speedup metric **overestimates efficiency** in the presence of **right-censored execution time** (i.e., when the program exceeds the time limit), as we discussed in Challenge C1.
>
> In fact, the time limits are exceeded very frequently in our evaluation, thanks to our high-quality reference solutions and challenging test cases. For example, **even for GPT-4, 16\% of its generated code exceeded the time limit** in our evaluation; for Mixtral 8x22B Instruct, 23\% of its generated code exceeded the time limit. Therefore, it is necessary to propose our eff metric to more rigorously handle right-censored execution time. We have added this experiment to Appendix B.
>
> |Rank|eff (ours)|speedup|
> |:-:|:--|:--|
> |1|GPT-4 Turbo|GPT-4 Turbo|
> |2|**GPT-4**|**Mixtral 8x22B Instruct**|
> |3|Llama 3 70B Instruct|Llama 3 70B Instruct|
> |4|**Mixtral 8x22B Instruct**|**GPT-4**|
> |5|Claude 3 Opus|Claude 3 Opus|
> |6|Phind Code Llama V2|Phind Code Llama V2|
> |7|**Claude 3 Haiku**|**ChatGPT**|
> |8|**ChatGPT**|**Claude 3 Haiku**|
> |9|Claude 3 Sonnet|Claude 3 Sonnet|
> |10|Llama 3 8B Instruct|Llama 3 8B Instruct|
> |11|**Code Llama 34B Python**|**Mixtral 8x7B Instruct**|
> |12|**Mixtral 8x7B Instruct**|**Code Llama 34B Python**|
>
> > *W2: Not evaluated with a prompt that asks for an efficient solution, I suspect that a better prompt would make this entire suite too weak for being a competitive benchmark.*
>
> Thanks for this interesting suggestion.  While thoroughly studying how to enhance efficiency is beyond the scope of this paper and probably requires another investigation, here we provide two experiments of prompt engineering as follows to better answer the reviewer's question.
>
> **Experiment I.** We use the two strongest open-source LLMs Llama 3 70B Instruct and Mixtral 8x22B Instruct from our Table 3 (main results). We ask the LLM to generate "the most efficient algorithm" and use greedy decoding to obtain outputs. The evaluation results are presented below.
>
> |Llama 3 70B Instruct|eff@1|pass@1|
> |:-:|:-:|:-:|
> |Basic prompt|0.421|0.746|
> |"most efficient algorithm"|0.418|0.746|
>
> |Mixtral 8x22B Instruct|eff@1|pass@1|
> |:-:|:-:|:-:|
> |Basic prompt|0.408|0.746|
> |"most efficient algorithm"|0.426|0.732|
>
> The results show that simple prompt engineering cannot enhance code efficiency much, even for such strong LLMs. The eff@1 of Mixtral 8x22B Instruct increases a little bit but is still far from the expert level. Interestingly, the eff@1 of Llama 3 70B Instruct even drops a little bit while the pass@1 does not change. This suggests that even such strong LLMs lack a good understanding of code efficiency.
>
> **Experiment II.** We believe that prompting alone does not fully address the code efficiency issue because this is essentially a **fundamental capability limitation** of existing LLMs---the efficient algorithm can be highly non-trivial so that existing LLMs do not understand well. To demonstrate this, we have conducted a case study for a hard problem (\#36) under the following two settings: (i) **Self-Refine** prompting (Madaan et al., 2023) and (ii) **revealing the efficient algorithm** in the prompt. (The efficient algorithm for \#36 is a dynamic programming over digits in $O(\log n)$ time; see Appendix E.2). Since the outputs are too long, the complete outputs are attached **at the end of our response**. We can see that, **no matter whether we use (i) or (ii)**, the generated code is an inefficient algorithms in $O(n)$ time, which is far from the optimal $O(\log n)$ time complexity.
>
> We have added these two interesting experiments to Appendix B.

---

> > ### Comment · Reviewer_xNv8 · 2024-11-26
> >
> > Thank you, this is very helpful.

---

> ### Author Response · Authors · 2024-11-23
> **Response to Reviewer xNv8 (Part 2/4)**
>
> > *Q1: I understand the theoretical importance of Rao-Blackwellization in this case, but what is the empirical effect?*
>
> Thank you for the great question. This is due to a statistical consideration.
>
> As LLMs' autoregressive generation is a random process, the eff@k metric is actually a random variable. If eff@k has a high variance, then the accuracy of the observed eff@k will not be guaranteed. To theoretically guarantee an accurate estimation, we need to reduce the variance of the eff@k estimator. Therefore, we employ Rao--Blackwellization to reduce the variance of the eff@k estimator. As shown in Theorem 1, if we use n code samples to estimate eff@k, then the variance of the Rao--Blackwellized estimator will be k/n times the variance of the original estimator. Hence, increasing n reduces the variance of the eff@k estimator and thus gives a more accurate estimate of LLMs' capabilities.
>
> To demonstrate that Rao--Blackwellization does reduce the variance of the eff@k estimator, we empirically compute the standard deviation (i.e., sqrt of variance) of the vanilla eff@k estimator and our Rao--Blackwellized eff@k estimator (RB) using the Llama 3 70B Instruct model. Results are shown in the table below. We can see that the RB estimator achieves much lower standard deviation than the vanilla estimator.
> Therefore, our Rao--Blackwellized estimator empirically ensures a more accurate estimation of eff@k. We have added this experiment to Appendix B.
>
> |Estimator|k=1|k=10|
> |:--|:-:|:-:|
> |Vanilla|0.20|0.25|
> |**RB**|**0.02**|**0.08**|
>
> > *Q2: Effibench presented an efficiency-benchmark with 1,000 coding problems, where ach problem is paired with an executable human-written canonical solution, which obtains the SOTA efficiency on the LeetCode solution leaderboard. I would not classify it as a sporadic attempt, what makes you view it as such?*
>
> Thank you for pointing out this. By "sporadic" we just meant that the number of efficiency benchmarks (e.g., EffiBench, ENAMEL) is much fewer than the number of correctness benchmarks (e.g., HumanEval, HumanEval+, MBPP, APPS, DS-1000, MultiPL-E, CoNaLa, and many more). We agree with the reviewer that EffiBench is a nice large-scale benchmark for efficiency evaluation.
>
> To clarify this, we have revised "sporadic attempts" into "a few benchmarks" in Section 1.
>
> > *Q3: Does it make sense to also consider space complexity in your benchmarks? For example, ... a suggested solution uses two pointers, has the same time complexity, and reduces space complexity from O(n) to O(1).*
>
> Thank you for the great suggestion. While our current benchmark focuses on evaluating time efficiency, we believe that evaluating the space efficiency would be a very interesting and important future research direction. For example, EffiBench is a time--space joint evaluation benchmark for LLM-generated code. A potential challenge is how to evaluate the time--space trade-off. Since many time-efficient algorithms trade space for time (e.g., dynamic programming), a space-optimal algorithm may be less time-efficient, and vice versa. We may need to provide different reference solutions for time evaluation and space evaluation. We have added a discussion to Appendix D.
>
> Regarding your suggested solution, we chose the set-based solution because it is more time-efficient in Python (although it is a bit less space-efficient than the two-pointer solution). For instance, for $n=600$, our set-based solution runs in **6 ms** while the two-pointer solution runs in **29 ms**. As we discussed in Section 3.2, we care about not only time complexity but also **implementation optimization**. The set in Python is very efficient, so the set-based solution runs faster than the two-pointer solution in Python. This example also demonstrates **why we employ human expert** to develop this benchmark. (On the other hand, if our goal were to evaluate space efficiency, then the two-pointer solution would be a better reference solution.)
>
> ------------------------
>
> References:
>
> - Amdahl. "Validity of the single processor approach to achieving large scale computing capabilities." Proceedings of the April 18-20, 1967, Spring Joint Computer Conference. 1967.
> - Madaan et al. Self-Refine: Iterative refinement with self-feedback. 2023.
> - Niu et al. On evaluating the efficiency of source code generated by LLMs. 2024.

---

> > ### Comment · Reviewer_xNv8 · 2024-11-26
> >
> > excellent response, thank you

---

> ### Author Response · Authors · 2024-11-23
> **Response to Reviewer xNv8 (Part 3/4)**
>
> The following are results for Experiment II in W2.
>
> ## (i) Self-Refine prompting (Madaan et al., 2023)
>
> Following Niu et al. (2024), we ask the LLM to "give a potential strategy to improve the efficiency of the code" and finally to "give the optimized version of the same code with the strategy mentioned above." The code before and after Self-Refine is attached below for your reference. Before Self-Refine, both Llama 3 70B Instruct and Mixtral 8x22B Instruct generate the brute-force algorithm that runs in $O(n)$ time. After Self-Refine, both LLMs apply a small improvement to the generated code, but the time complexity is still $O(n)$, significantly slower than the efficient algorithm that runs in $O(\log n)$ time. This suggests that both LLMs are unable to come up with the efficient algorithm on their own.
>
> ------------------------
>
> Initial code by Llama 3 70B Instruct:
>
> ```python
> def fizz_buzz(n: int) -> int:
>     """Return the number of times the digit 7 appears in integers less than n which are divisible by 11 or 13."""
>     count = 0
>     for i in range(n):
>         if i % 11 == 0 or i % 13 == 0:
>             count += str(i).count('7')
>     return count
> ```
>
> Refined code by Llama 3 70B Instruct:
>
> ```python
> import math
>
> def fizz_buzz(n: int) -> int:
>     """Return the number of times the digit 7 appears in integers less than n which are divisible by 11 or 13."""
>     return sum(str(i).count('7') for i in range(n) if math.gcd(i, 11 * 13) in (11, 13))
> ```
>
> Initial code by Mixtral 8x22B Instruct:
>
> ```python
> def fizz_buzz(n: int) -> int:
>     count = 0
>     for i in range(11, n, 1):
>         if i % 11 == 0 or i % 13 == 0:
>             count += str(i).count('7')
>     return count
> ```
>
> Refined code by Mixtral 8x22B Instruct:
>
> ```python
> def fizz_buzz(n: int) -> int:
>     count = 0
>     for i in range(11, n, 11):
>         count += str(i).count('7')
>     for i in range(13, n, 13):
>         if i % 11 != 0:  # to avoid double counting numbers that are divisible by both 11 and 13
>             count += str(i).count('7')
>     return count
> ```

---

> ### Author Response · Authors · 2024-11-23
> **Response to Reviewer xNv8 (Part 4/4)**
>
> The following are results for Experiment II in W2.
>
> ## (ii) Revealing the efficient algorithm in the prompt
>
> In (i), we have shown that LLMs are unable to come up with the dynamic programming algorithm on their own. To further investigate whether LLMs can generate the dynamic programming algorithm here, we explicitly **reveal the efficient algorithm** in the prompt. Specifically, we use "Complete the following function with a dynamic programming over digits" for both LLMs. The raw outputs are attached below for your reference. Llama 3 70B Instruct fails to use dynamic programming; Mixtral 8x22B Instruct uses a trivial dynamic programming that is **not over digits**. We can see that both LLMs still generate inefficient algorithms that run in $O(n)$ time even though we have revealed the efficient $O(\log n)$ algorithm to them. This suggests that they do not have a good understanding of this non-trivial dynamic programming.
>
> From the two case studies above, we can see that the code inefficiency is essentially a **capability limitation** of existing LLMs, and therefore we do not expect it to be easily addressed merely via prompting techniques. This is also our **core motivation** for developing this benchmark to **compare LLMs with human experts**. While how to address this capability limitation is beyond the scope of this work, we hope that our work will motivate LLM developers pay more attention to this limitation and try to address it in future work.
>
> --------
>
> Output of Llama 3 70B Instruct:
>
> ~~~
> Here is the Python function that uses dynamic programming over digits to solve the problem:
> ```python
> def fizz_buzz(n: int):
>     count = 0
>     for i in range(n):
>         if i % 11 == 0 or i % 13 == 0:
>             count += str(i).count('7')
>     return count
> ```
> Here's an explanation of the code:
>
> 1. We initialize a variable `count` to 0, which will store the total count of digit 7 appearances.
> 2. We iterate over the range of integers from 0 to `n-1` (since we want to consider integers less than `n`).
> 3. For each integer `i` in the range, we check if it is divisible by 11 or 13 using the modulo operator (`%`). If it is, we proceed to the next step.
> 4. We convert the integer `i` to a string using the `str()` function, and then use the `count()` method to count the number of occurrences of the digit '7' in the string.
> 5. We add the count of digit 7 appearances in the current integer to the overall `count` variable.
> 6. Finally, we return the total `count` at the end of the function.
>
> This function has a time complexity of O(n), where n is the input integer, since we iterate over the range of integers from 0 to `n-1`. The space complexity is O(1) since we only use a single variable to store the count.
>
> You can test the function with the provided examples to verify that it produces the correct output:
> ```python
> print(fizz_buzz(50))  # 0
> print(fizz_buzz(78))  # 2
> print(fizz_buzz(79))  # 3
> ```
> ~~~
>
> Output of Mixtral 8x22B Instruct:
>
> ~~~
> Here is a Python function that uses dynamic programming to solve the problem:
>
> ```python
> def fizz_buzz(n: int):
>     """Return the number of times the digit 7 appears in integers less than n which are divisible by 11 or 13.
>     """
>     # Initialize a list to store the count of 7s in each number
>     dp = [0] * n
>
>     # Base case: the number of 7s in 0 is 0
>     dp[0] = 0
>
>     # Iterate over the range of numbers
>     for i in range(1, n):
>         # If the current number is divisible by 11 or 13
>         if i % 11 == 0 or i % 13 == 0:
>             # If the last digit of the current number is 7
>             if i % 10 == 7:
>                 # Add 1 to the count of 7s in the current number
>                 dp[i] = dp[i - 1] + 1
>             else:
>                 # Otherwise, the count of 7s in the current number is the same as the previous number
>                 dp[i] = dp[i - 1]
>         else:
>             # If the current number is not divisible by 11 or 13, the count of 7s is the same as the previous number
>             dp[i] = dp[i - 1]
>
>     # Return the count of 7s in numbers less than n
>     return dp[n - 1]
> ```
>
> This function works by iterating over the range of numbers from 1 to n-1 and checking if each number is divisible by 11 or 13. If it is, the function checks if the last digit of the number is 7. If it is, the function adds 1 to the count of 7s in the current number. Otherwise, the count of 7s in the current number is the same as the previous number. The function then returns the count of 7s in numbers less than n.
>
> Here are the test cases:
>
> ```python
> print(fizz_buzz(50))  # Output: 0
> print(fizz_buzz(78))  # Output: 2
> print(fizz_buzz(79))  # Output: 3
> ```
> ~~~

---

### Author Response · Authors · 2024-11-23
**General Response**

We sincerely thank all reviewers for their valuable time and thoughtful reviews. We appreciate reviewers' recognition that efficiency evaluation is an important problem (t6vW), that the benchmark provides high-quality reference solutions (xNv8, t6vW, 1vcL) and test cases (xNv8) are useful for future research (t6vW), that the evaluation metric is rigorously designed (xNv8, 2JoC), that the evaluation is comprehensive (t6vW) and insightful (2JoC, 1vcL), and that this paper has a clear presentation (2JoC, 1vcL) and is easy to follow (2JoC, 1vcL). We also thank reviewers for their constructive comments, which have inspired us to further improve the quality of the paper and will lead to interesting future research directions.

Regarding why we craft this benchmark by human expert and regarding the size of our benchmark, we provide our response as follows.

Firstly, we believe that crafting the benchmark by **human expert is necessary** to develop a **high-standard and rigorous** benchmark. For example, as shown in Table 2, compared with our expert solutions, HumanEval canonical solutions achieve only eff@1=0.455, and HumanEvalPlus canonical solutions achieve only eff@1=0.513. This shows that their canonical solutions are **far less efficient than our expert-written reference solutions**.

  - (i) We did not use problems/solutions from **online judges** (like LeetCode or Codeforces) because their **public solutions are already in LLMs' pretraining corpuses**. For example, DeepMind's AlphaCode [1] has been trained on many online judges including Codeforces, CodeChef, HackerEarth, AtCoder, and Aizu. If we evaluate LLMs on these public online judges, then the evaluation results may fail to reflect the LLMs’ true capabilities due to test set leakage / memorization.
  - (ii) We did not **crowd-source** the benchmark because otherwise it would be **hard to guarantee the quality** of the benchmark. For example, MBPP [2] is a popular crowd-sourced benchmark, but it is known to be easier than HumanEval [3].
  - (iii) We did not use **LLM-generate reference solutions** because LLM-generated code are still far from expert-level efficiency, as demonstrated in our evaluation.

Secondly, despite the size of the benchmark, **our 142 problems has already revealed the limited capability** of all the 30 LLMs in generating efficient code. In particular, our benchmark shows that even the strongest LLM (GPT-4 Turbo) is still far from generating expert-level efficient code (eff@1 below 0.5). We hope our findings and benchmark will help LLM developers to realize this critical issue and further inspire them to develop stronger LLM code generators. The effectiveness of our benchmark is because **our human expert has carefully verified the comprehensiveness** of the problemset:

  - (i) As shown in Figure 2, our benchmark problems **have diverse difficulties**. For example, 75 seemingly easy problems require non-trivial implementation optimization, and 20 hard problems require advanced algorithms.
  - (ii) As discussed in Section 3.2, our problemset **covers a wide range of algorithmic knowledge** (including data structures, dynamic programming, and automata) and **a wide range of mathematical knowledge** (including linear algebra, combinatorics, and number theory).

That said, we still agree with the reviewers that addressing scalablity of benchmark development is an important future direction. A possible solution is to **collaborate with private programming competitions** whose solutions are not publicly available. Nevertheless, we believe that this will be an issue *only after* LLMs can generate efficient code on our benchmark but might not for now. We have added a discussion to Appendix D.

For other questions, we provide our response below and have revised our paper accordingly. We warmly invite the reviewers for further discussion.

-------------

References:

- [1] Li et al. Competition-level code generation with AlphaCode. Science. 378(6624):1092-
1097, 2022.
- [2] Austin et al. Program synthesis with large language models. Google Research, 2021.
- [3] Rozière et al. Code Llama: Open foundation models for code. Meta AI, 2023.

---

### Meta-Review · Area_Chair_mTtd · 2024-12-19

**Metareview:**

The paper introduces a new metric (and Monte Carlo estimator) for the expected performance of a program sampled from a language model. They use this new theoretical advance to then do a very broad evaluation of language models, and a very narrow evaluation of programming problems (the primary weakness). But this weakness is mostly because they need to hand write efficient solutions to various problems in order to normalize performance, so it is understandable. I think that both the problem as well as the theoretical result are important for the community, hence support acceptance.

**Additional Comments On Reviewer Discussion:**

Those who responded were generally happy with the rebuttal, such as the discussion of how programming contests could provide a good source of new test cases.

---

### Decision · Program_Chairs · 2025-01-22

Accept (Poster)